# The RNA-binding protein Puf5 and the HMGB protein Ixr1 regulate cell cycle-specific expression of *CLB1* and *CLB2* in *Saccharomyces cerevisiae*

Megumi Sato[1,2⊙], Varsha Rana[1,3⊙], Yasuyuki Suda[1,4], Tomoaki Mizuno[1], Kenji Irie[1]*

1 Laboratory of Molecular Cell Biology, Institute of Medicine, University of Tsukuba, Tsukuba, Japan, 2 Doctoral Program in Medical Sciences, Graduate School of Comprehensive Human Sciences, University of Tsukuba, Tsukuba, Japan, 3 Doctoral Program in Human Biology, Graduate School of Comprehensive Human Sciences, University of Tsukuba, Tsukuba, Japan, 4 Live Cell Super-Resolution Imaging Research Team, RIKEN Center for Advanced Photonics, Wako, Saitama, Japan

⊙ These authors contributed equally to this work.
* kirie@md.tsukuba.ac.jp

**Data Availability Statement:** All relevant data are within the paper and its Supporting Information files.

## Abstract

Clb1 and Clb2 are functionally redundant B-type cyclins, and the *clb1Δ clb2Δ* double mutant is lethal. In normal mitotic growth, Clb2 plays the central role in the G2-M progression. We previously demonstrated that the RNA-binding protein Puf5 positively regulates *CLB1* expression by downregulating expression of the repressor Ixr1. The decreased expression of *CLB1* by the *puf5Δ* mutation caused a severe growth defect of the *puf5Δ clb2Δ* double mutant. On the contrary, *CLB2* expression was unchanged between wild-type strain and *puf5Δ* mutant in unsynchronized cultures, and the *puf5Δ clb1Δ* double mutant did not show growth retardation. Therefore, we assumed that *CLB1* is the main target of Puf5 in the previous study. However, considering that *CLB1* and *CLB2* reportedly undergo a similar expression pattern during the cell cycle, we re-examined *CLB2* expression in the *puf5Δ* mutant in cell cycle-synchronized cultures and found that *CLB2* expression was decreased in the *puf5Δ* mutant strain. Deletion of *IXR1* restored the decreased expression of *CLB2* caused by the *puf5Δ* mutation. Moreover, we clarified that the decreased expression of *CLB2* caused by the *puf5Δ* mutation resulted in the growth defect in the S-phase cyclin deficient condition: the *puf5Δ clb1Δ clb5Δ clb6Δ* quadruple mutant grew worse than *clb1Δ clb5Δ clb6Δ* triple mutant, and the slow growth of the *puf5Δ clb1Δ clb5Δ clb6Δ* quadruple mutant was suppressed by *CLB2* overexpression. Moreover, the *ixr1Δ* mutation is known to be synthetically lethal with deletion of the *DUN1* gene encoding the checkpoint kinase. We found that the *clb2Δ* mutation restored the lethality of *ixr1Δ dun1Δ* double mutant. Our results suggest that Puf5 and Ixr1 regulate the cell cycle-specific expression of both *CLB1* and *CLB2*, that Clb5 and Clb6 have overlapping roles with Clb1 and Clb2, and that the regulation of *CLB1* and *CLB2* expression by Puf5 and Ixr1 is related to the function of Dun1 kinase.

**Funding:** This research was supported by Japan Society for the Promotion of Science (JSPS) KAKENHI Grant Number 22K06074 (to KI). The funders had no role in study design, data collection and analysis, decision to publish, or preparation of the manuscript.

**Competing interests:** The authors have declared that no competing interests exist.

## Introduction

The eukaryotic cell cycle is a stringently regulated biological process that controls cellular growth and division, and maintains genomic integrity. The precise timing and coordination of the cell cycle are controlled by the periodic activity of the complex of cyclin-dependent kinases (Cdks) and cyclins [1]. In *Saccharomyces cerevisiae*, commonly known as the budding yeast, the cell cycle is primarily driven by a single Cdk called Cdc28 [1]. Nine periodically expressed cyclin proteins, broadly categorized into G1 cyclins and B-type cyclins, activate Cdc28 at different phases. The G1 cyclins include Cln1, Cln2, and Cln3, which are essential for surpassing START and transitioning the cell cycle from G1 to S-phase. In contrast, the B-type cyclins encompass six different cyclins viz., Clb1, Clb2, Clb3, Clb4, Clb5, and Clb6. These cyclins are further grouped into three unique pairs according to homology and transcriptional regulation, Clb1/Clb2, Clb3/Clb4, and Clb5/Clb6. The transcriptional induction of these pairs occurs in three distinct waves at different phases of the cell cycle [1–3]. For instance, *CLB5/CLB6* peaks just before the initiation of the cell cycle or S-phase and promotes efficient initiation of DNA replication. *CLB3/CLB4* peaks when the DNA replication is completed and promotes mitotic spindle formation at the G2/M-phase transition. *CLB1/CLB2* peaks during the transition from G2 to M-phase and are essential for mitotic spindle elongation, spindle pole body separation, and mitotic exit [4–7]. The functional variation of the cyclins is attributed to the timing of the accumulation of cyclins. However, previous studies have shown plasticity and functional overlap amongst the cyclins expressed in different stages of the cell cycle. For example, in the absence of S-phase cyclins Clb5 and Clb6, other cyclins are able to facilitate origin firing and DNA replication, albeit with a delay [8,9]. In contrast, mitotic exit can only be achieved by Clb2, not by Clb5 [7,8]. Therefore, B-type cyclins somehow show functional redundancy, but some crucial functions seem to be specific.

Among the B-type cyclins, the Clb2 cyclin is the major cyclin and is critical for mitotic entry [4,5]. During the G2/M-phase, the transcription of G2/M cluster genes including *CLB1* and *CLB2* is induced by the transcriptional activator Mcm1-Fkh2-Ndd1 complex [2,10–13]. Interestingly, Clb2 acts in a positive feedback loop; the Clb2/Cdc28 kinase complex phosphorylates Fkh2 and Ndd1 during the G2 phase of the cell cycle, further promoting assembly and activation of the Mcm1-Fkh2-Ndd1 complex [11,13,14]. In addition, Clb2 also cooperates with the other G2/M cluster genes to inhibit Swi4, a component of the SBF transcription complex, and negatively regulates the expression of the SBF-regulated genes, including the G1 cyclins *CLN1* and *CLN2* [14]. This further reiterates the significance of G2/M cyclins in the regulation of not only mitosis but also other stages of the cell cycle. Thus, considering the importance of G2/M cyclins, elucidating G2/M regulators could further help in the understanding of the cell cycle progression.

RNA-binding proteins play an integral part of the transcriptional and post-transcriptional control machinery, thereby providing additional control over the gene expression. In the context of cell cycle regulation, RNA-binding proteins have emerged as potential regulators of cyclins and other cell cycle-related transcripts. Puf5 (also known as Mpt5 or Htr1) is one such RNA-binding protein that has been reported to be essential for the G2/M phase transition of the cell cycle at high temperatures [15,16]. Puf5 belongs to the PUF (Pumilio and FBF) family, which is one of the highly evolutionary conserved families of RNA-binding proteins. Puf5 binds to the short motifs in the 3'UTR of an mRNA via its Pumilio homology domain and regulates the expression, stability, localization, and efficiency of translation of the bound transcript [17–19], contributing to cell growth by regulating diverse phenomena including cell wall integrity [20,21], maintenance of longevity [22], and mating type switching [23]. Puf5 has been reported to bind to more than 1000 mRNAs, which constitute approximately 16% of the yeast

transcriptome [24], but physiologically important targets of Puf5 have not been elucidated well.

Recently, we reported that Puf5 positively regulates the expression of *CLB1* via the HMGB (High Mobility Group Box B) protein Ixr1 specifically in G2/M-phase. Mechanistically, the binding of Puf5 to the 3'UTR of *IXR1* mRNA suppresses its expression, which subsequently increases the expression of *CLB1* [25]. In this study, we further elaborated on the role of Puf5-Ixr1 in the regulation of cell cycle and cell growth. We found that under cell cycle-synchronized conditions, Puf5 positively regulates the expression of not only *CLB1* but also *CLB2*. Ixr1 also acts as a negative regulator of the *CLB2* expression. Moreover, we found that the deletion of Puf5 causes a synthetic growth defect with *clb5Δ* mutation and the *clb5Δ clb6Δ* double mutation, which resulted from a decreased expression of *CLB2* caused by the *puf5Δ* mutation. We also found that Clb2 overexpression was able to compensate for the absence of G2/M-phase and S-phase cyclins. In addition, the proper level of *CLB2* expression under the control by Ixr1 was necessary for cell growth under the DNA-damage uninducible *dun1Δ* mutation condition. Altogether, our study suggests that Puf5 and Ixr1 maintain appropriate expression of *CLB1* and *CLB2*, thereby contributing to the specific utilization of Clb1/2 and Clb5/6 throughout the cell cycle.

## Results

### Expression of not only *CLB1* but also *CLB2* was decreased in the *puf5Δ* mutant

*CLB1* and *CLB2* are functionally redundant B-type cyclins, and *clb1Δ clb2Δ* double mutant is lethal. In normal mitotic growth, Clb2 plays the central role for the G2-M progression [4,5]. In our previous paper [25], we showed that Puf5 positively regulates *CLB1* expression by binding to *IXR1* mRNA and negatively regulating its expression. *CLB1* expression was reduced in the *puf5Δ* mutant compared to the wild-type strain (Fig 1A). Referring to the data that the *puf5Δ* single mutant grew similarly to the wild-type strain (Fig 1B), this regulation by Puf5 solely does not have a phenotypic effect. However, when Clb2 was absent, the *puf5Δ* mutation caused a severe growth defect (Fig 1B, *puf5Δ clb2Δ*), indicating that Puf5 ensures *CLB1* expression in the case where the function of Clb2 is lost.

On the contrary, although *CLB1* and *CLB2* are under the same expressional control machinery, *CLB2* expression was unchanged between wild-type strain and the *puf5Δ* mutant in unsynchronized cultures (Fig 1A). In addition, the *puf5Δ clb1Δ* double mutant grew similarly to the wild-type strain and the *puf5Δ* single mutant (Fig 1B). Although this result suggests that Puf5 is involved in the regulation of *CLB1* but not *CLB2* expression, it seems curious that regulation by Puf5 differs between the same cluster genes, *CLB1* and *CLB2*. Therefore, we re-examined *CLB2* expression in the *puf5Δ* mutant in cell cycle-synchronized cultures. In this experiment, we first arrested the cell cycle in the G1-phase using α-factor, then released, and collected samples every 10 minutes. The *bar1Δ* mutation background was utilized for inhibiting degradation of α-factor. The RNA levels of *RNR1* and *SIC1* were examined as cell cycle markers of S-phase and late M-phase, respectively. In the *bar1Δ* mutant, *RNR1* expression was highest at 40 minutes (Fig 2A), and *SIC1* expression was peaked at 80 minutes (Fig 2B). Regarding the *bar1Δ puf5Δ* double mutant, *RNR1* expression was induced the most at 50 minutes (Fig 2A), and *SIC1* at 110 minutes (Fig 2B), implying that cell cycle was delayed in the *bar1Δ puf5Δ* double mutant, especially in the G2-phase. Consistent with the data shown in Fig 1A, *CLB1* expression was decreased in the *bar1Δ puf5Δ* double mutant compared to the *bar1Δ* mutant (Fig 2C). Interestingly, in cell cycle-synchronized cultures, a decrease in *CLB2* expression was observed in the *puf5Δ* mutant (Fig 2D). However, the decrease in *CLB2*

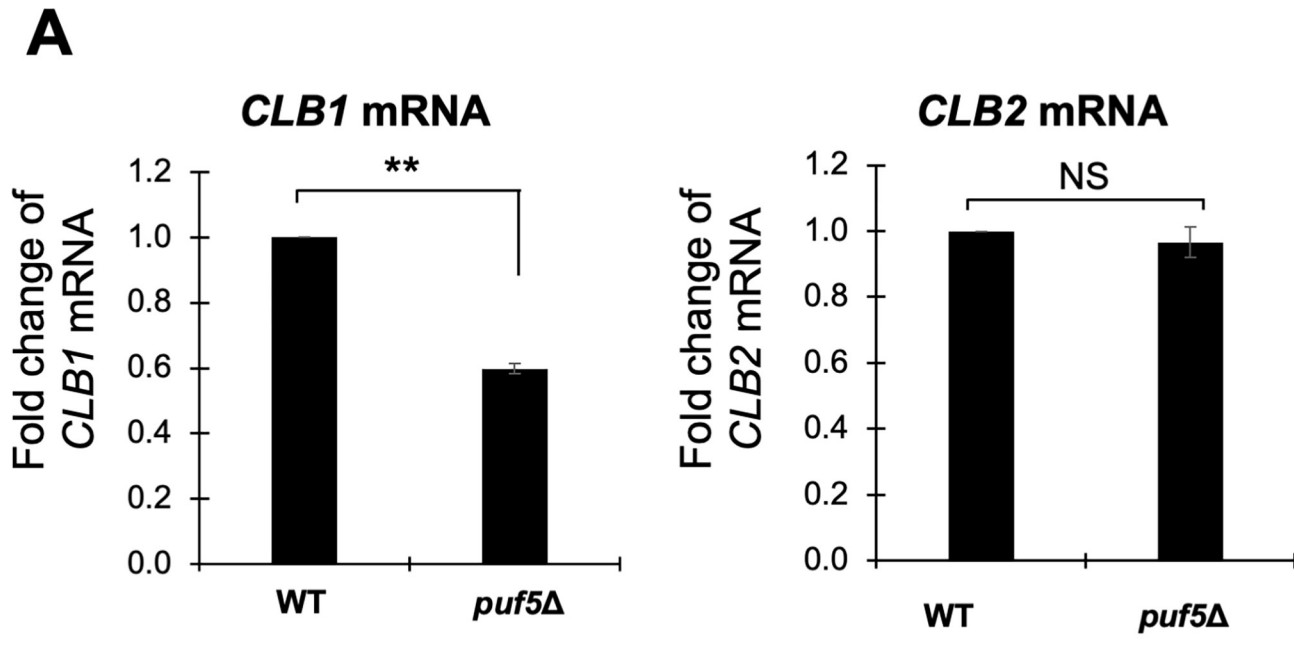

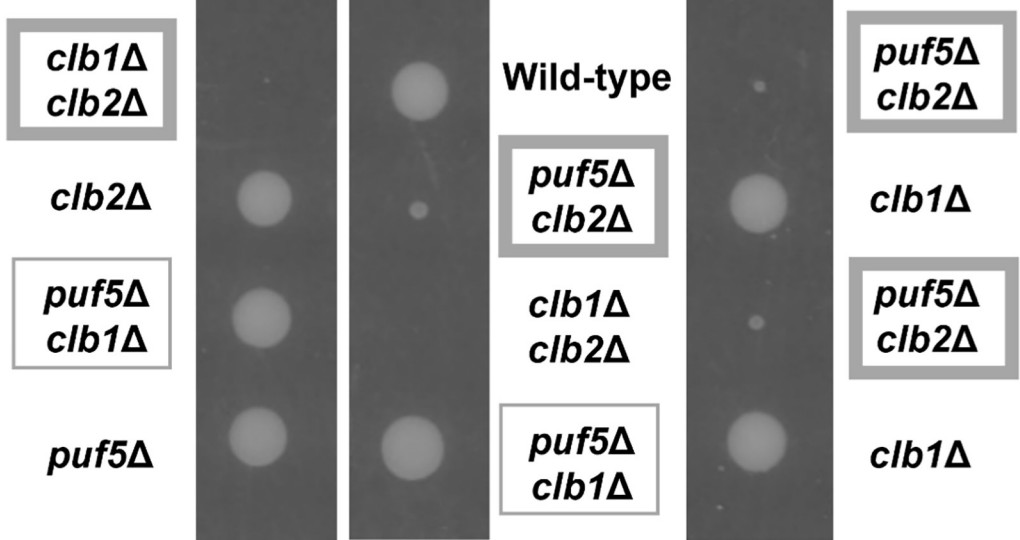

**Fig 1. Puf5 positively regulates the expression of *CLB1* and *CLB2*.** (A) The mRNA levels of *CLB1* and *CLB2* and in the wild-type strain and the *puf5Δ* mutant. The cells were cultured in a YPD medium at 28˚C until the log phase. The *CLB* mRNA levels were quantified by qRT-PCR analysis, and the relative mRNA levels were calculated using the *SCR1* reference gene. The data shows the mean ± SE (n = 3) of the fold change of *CLB1* and *CLB2* mRNA levels relative to the mRNA level in the wild-type strain. **P < 0.01 indicates statistical significance, and NS does no significant change. (B) The tetrad analysis of the strains that are heterozygous for the alleles of *PUF5*, *CLB1*, and *CLB2*. The cells were sporulated, dissected on a YPD plate, and cultured at 30˚C for 3 days. The strains encircled in the wide frame show growth defect.

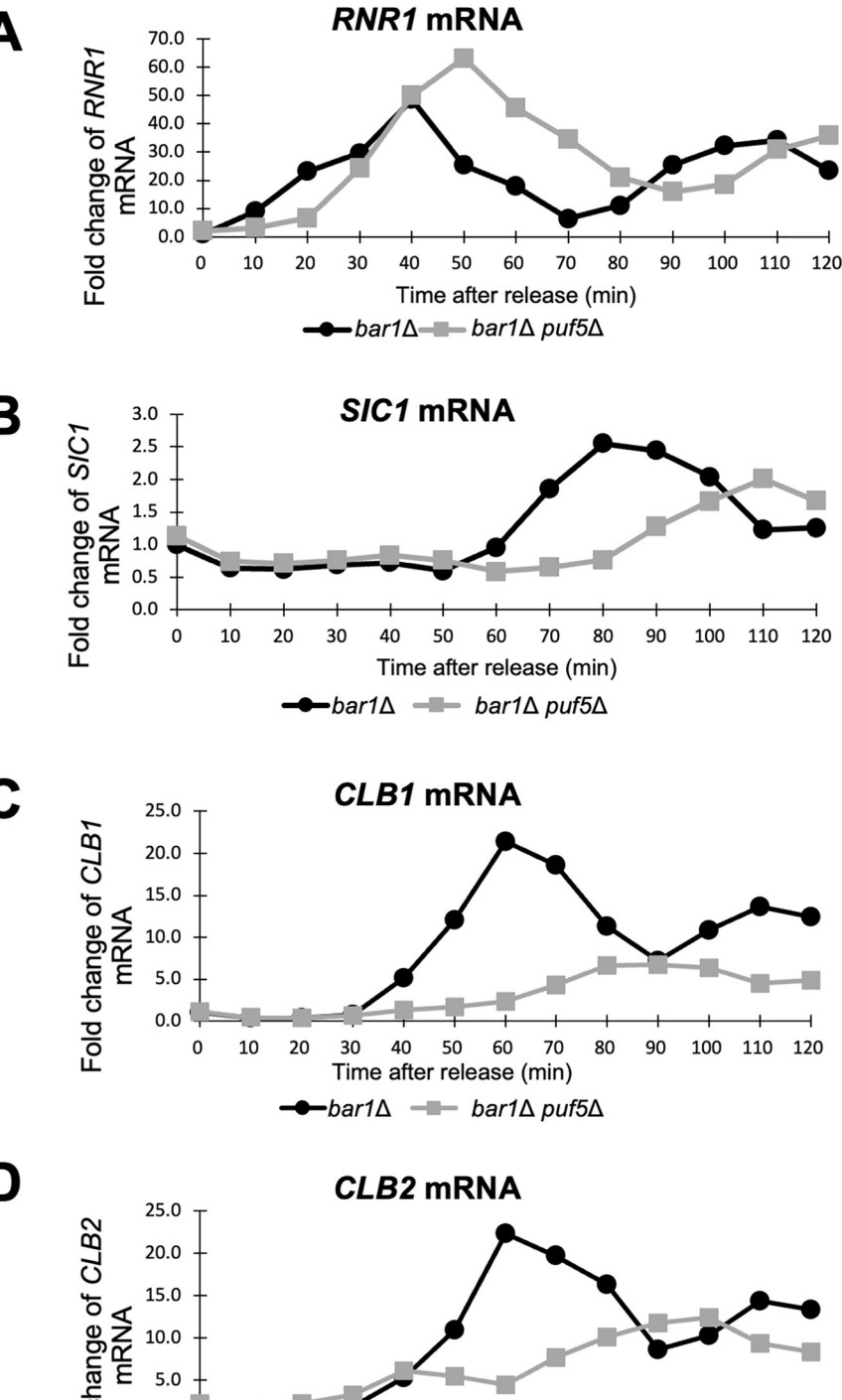

**Fig 2. The cell cycle-regulated expression of *CLB1* and *CLB2* was diminished in the *puf5Δ* mutant.** (A-D) The cell cycle-dependent mRNA levels of *RNR1*, *SIC1*, *CLB1*, and *CLB2* in the synchronized *bar1Δ* cell (black circle) and *bar1Δ puf5Δ* mutant (grey square). The levels of *RNR1* mRNA (A), *SIC1* mRNA (B), *CLB1* mRNA (C), and *CLB2* mRNA (D) were quantified by qRT-PCR analysis, and the relative mRNA levels were calculated using the *SCR1* reference gene. The vertical axis shows the fold change of mRNA level relative to that in the *bar1Δ* 0 min sample, and the horizontal

axis shows the time after release from G1-phase. This experiment was performed repeatedly (n = 2), and the representative data is shown.

expression in the *puf5Δ* mutant strain was milder than the decrease in *CLB1* expression (Fig 2C and 2D).

Next, we examined whether these decreases in the mRNA levels are also reflected in the protein levels. In this experiment, YCplac33-*CLB1*-3HA or YCplac33-*CLB2*-3HA plasmid was transformed into the *bar1Δ* mutant and the *bar1Δ puf5Δ* double mutant, and cell cycle-specific protein levels of Clb1-3HA and Clb2-3HA were investigated. Although the basal protein levels of Clb1-3HA were quite low, the levels were decreased in the *bar1Δ puf5Δ* double mutant compared to the control *bar1Δ* mutant at 60, 80, and 100 minutes (Fig 3A). The Clb2-HA protein levels were also lower in the *bar1Δ puf5Δ* double mutant than the *bar1Δ* mutant at 40 minutes, but, at 60, 80, 100 minutes, the Clb2-HA protein levels were similar between in the *bar1Δ puf5Δ* double mutant and the *bar1Δ* mutant (Fig 3B). The extent of the decrease in Clb2 protein levels was much milder than that of *CLB2* mRNA levels (Figs 2D and 3B), but it seems certain that at least the induction of Clb2 protein expression was delayed in the *puf5Δ* mutant. The reason why the decrease in Clb2 protein levels was so much milder than the decrease in *CLB2* mRNA levels is not clear, but may be due to differences in experimental conditions, or there may be another mechanism to refill protein levels. From these results, even though the effect on the Clb2 protein level was much weaker than that on the Clb1 protein level, it is suggested that Puf5 positively regulates the cell cycle specific expression of *CLB2* at both mRNA and protein levels in addition to *CLB1*.

Thus, the findings raise a key question whether the positive regulation of B-type cyclin genes by Puf5 is specific to the G2/M cyclin, *CLB1* and *CLB2*. To clarify the effect of Puf5 on other B-type cyclin genes, we additionally examined the expression of *CLB3*, *CLB4*, *CLB5* and *CLB6*. The *CLB3* and *CLB4* expression remained unchanged between the *bar1Δ* mutant and the *bar1Δ puf5Δ* double mutant in the synchronous culture (S1A and S1B Fig). Although the expression of *CLB5* and *CLB6* was induced more slowly and at a lower extent in the *bar1Δ puf5Δ* double mutant than in the *bar1Δ* mutant, the difference in their expression was far milder than that in *CLB1* and *CLB2* expression (Figs 2C and 2D, S1C and S1D). Thus, Puf5 appears to positively regulate *CLB1* and *CLB2* most significantly among B-type cyclin genes.

## Puf5 regulates *CLB2* expression via Ixr1

Puf5 is found to be involved in the regulation of *CLB2* expression as well as *CLB1* expression. We have reported that Ixr1, a HMGB protein functioning as a transcriptional repressor, is involved in the regulation of the expression of *CLB1* [25]. Therefore, we examined whether Ixr1 also contributes to the regulation of *CLB2* expression. As a result, *CLB2* expression, as well as *CLB1* expression, was increased in the *ixr1Δ* mutant compared to wild-type strain even in the asynchronous culture (Fig 4A and 4B). Moreover, the expression levels of *CLB1* and *CLB2* were higher in the *puf5Δ ixr1Δ* double mutant than in the *puf5Δ* single mutant (Fig 4A and 4B). These results indicate that Ixr1 also mediates the regulation of *CLB2* expression by Puf5. Next, we also examined cell cycle-specific expression of *CLB2* in the *ixr1Δ* mutant. When culturing three strains, the *bar1Δ* mutant, the *bar1Δ puf5Δ* double mutant, and the *bar1Δ puf5Δ ixr1Δ* triple mutant, the induction of cell phase markers, *RNR1* and *SIC1*, was delayed in the *bar1Δ puf5Δ* double mutant compared to the *bar1Δ* mutant, and this delay was recovered by the additional *ixr1Δ* mutation (Fig 5A and 5B). As reported in our previous paper [25], the expression of *CLB1* was decreased in the *bar1Δ puf5Δ* double mutant and restored in the *bar1Δ puf5Δ ixr1Δ* triple mutant strain (Fig 5C). Regarding the *CLB2*

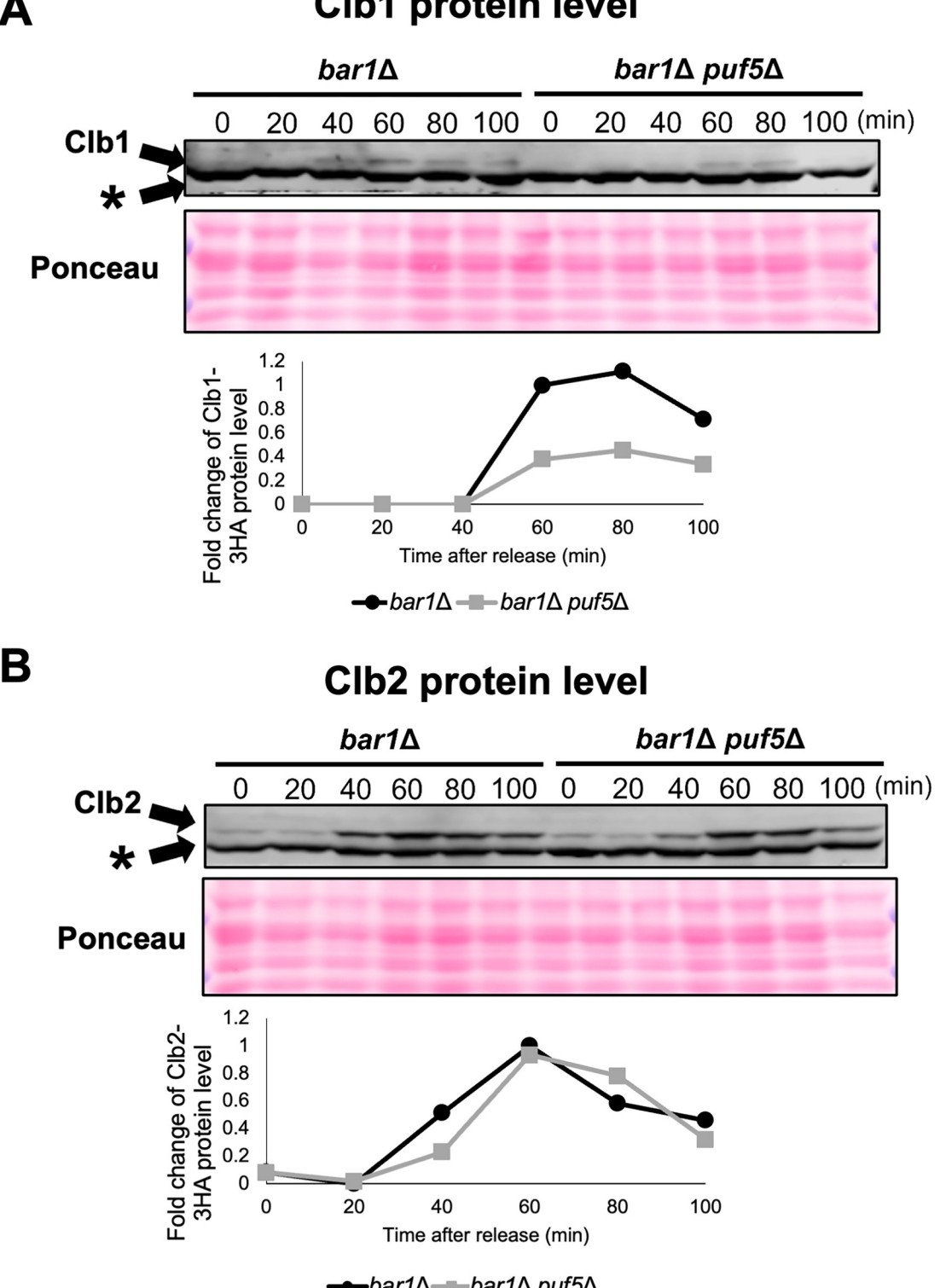

**Fig 3. Cell cycle-specific expression of Clb1 and Clb2 protein was also decreased in the *puf5Δ* mutant.** (A, B) The cell cycle-dependent Clb1 (A) and Clb2 (B) protein levels. The *bar1Δ* mutant and *bar1Δ puf5Δ* double mutant strains harboring YCplac33-*CLB1*-3HA-*CLB1* 3′UTR or YCplac33-*CLB2*-3HA-*CLB2* 3′UTR plasmid were synchronized by α-factor-induced G1 arrest. After releasing, samples were collected every 20 minutes, and proteins were extracted and immunoblotted with anti-HA antibody. This experiment was repeatedly performed (n = 2), and the representative blot image is presented. The band labeled as (*) presents

non-specific band of anti-HA antibody. The total protein abundance loaded was confirmed to be at the same level by Ponceau staining. The graphs show the fold change of the density of the Clb1-3HA or Clb2-3HA bands relative to that in the *bar1Δ* 60 minutes.

expression, the RNA level was decreased in the *bar1Δ puf5Δ* double mutant as shown in Fig 2D. When additionally deleted *IXR1*, the basal expression of *CLB2* was increased (Fig 5D). These results suggest that Ixr1 functions downstream of Puf5 and regulates the *CLB1* and *CLB2* expression.

Additionally, we investigated the cell cycle-dependent expression of *CLB2* in the *bar1Δ ixr1Δ* double mutant. In this experiment, samples were collected every 20 minutes after the cell cycle blocking and releasing. The S-phase marker *RNR1* expressed at most at the same time points in the *bar1Δ ixr1Δ* double mutant as the *bar1Δ* mutant at 20 minutes and 80 minutes, but the peaked expression of the second cell cycle was lower in the *bar1Δ ixr1Δ* double mutant (Fig 6A). The late M-phase marker *SIC1* showed a similar expressional pattern between the two strains examined (Fig 6B). As for *CLB1*, its expression was mildly higher in the *bar1Δ ixr1Δ* double mutant than in the *bar1Δ* mutant after starting to be induced at 20 minutes (Fig 6C). As for *CLB2*, the basal expression of *CLB2* at 0 minute was approximately 6-times higher in the *bar1Δ ixr1Δ* double mutant than in the *bar1Δ* mutant (Fig 6D). The *CLB2* expression peaked at 60 minutes in both strains, and the maximum induction level did not show a significant change between the two strains (Fig 6D). Altogether, although it is suggested that Ixr1 regulates not only *CLB1* but also *CLB2* expression in a downstream of Puf5, Ixr1 does not have strong effects on the cell cycle-dependent induction of *CLB2*. Rather, Ixr1 seems to control the basal expression of *CLB2* during the cell cycle.

## The effect of reduced expression of *CLB2* by the *puf5Δ* mutation was observed in the *clb1Δ clb5Δ clb6Δ* triple mutant strains

In our previous study, we observed that the reduced *CLB1* expression in the *puf5Δ* mutant resulted in the growth defect of the *puf5Δ clb2Δ* double mutant [25] (Fig 1B). On the other hand, the *puf5Δ clb1Δ* double mutant grew similarly to the wild-type strain (Fig 1B). Why did the *puf5Δ clb1Δ* double mutant grow as well as the wild-type strain, even though *CLB2* expression was reduced by the *puf5Δ* mutation? We hypothesized that the other B-type cyclins, Clb3, Clb4, Clb5 and Clb6, fulfill the function of Clb1 and Clb2. To verify this hypothesis, we investigated the genetical interaction between *PUF5* and each of the four cyclin genes. We found that the *puf5Δ clb5Δ* double mutant and the *puf5Δ clb5Δ clb6Δ* triple mutant exhibited markedly slow growth than the *puf5Δ* mutant (Fig 7A), while the *puf5Δ clb3Δ clb4Δ* triple mutant just showed only a slight growth retardation (S2 Fig). This observation suggests that the reduced *CLB2* expression by the *puf5Δ* mutation has a marked physiological importance in the background where S-phase cyclins are absent. Considering that Puf5 positively regulates both *CLB1* and *CLB2*, we next examined which regulation is more important, *CLB1* or *CLB2*. The tetrad analysis revealed that *clb1Δ clb5Δ clb6Δ* triple mutant grew slightly slower than wild-type strain, and that the *puf5Δ clb1Δ clb5Δ clb6Δ* quadruple mutant showed a severe growth defect or sometimes lethality (Fig 7B). These results imply that the decreased *CLB2* expression level caused by the *puf5Δ* mutation is physiologically important in the *clb5Δ clb6Δ* double mutation background. Next, we examined whether the cell cycle-specific expression of *CLB2* was actually decreased in the *puf5Δ clb1Δ clb5Δ clb6Δ* quadruple mutant. For synchronously culturing these strains, we utilized the *BAR1* positive background and used 100 times higher concentration of α-factor as reported previously [26]. It is confirmed that *CLB2* expression was induced slower in the *puf5Δ clb1Δ clb5Δ clb6Δ* quadruple mutant than in the *clb1Δ clb5Δ*

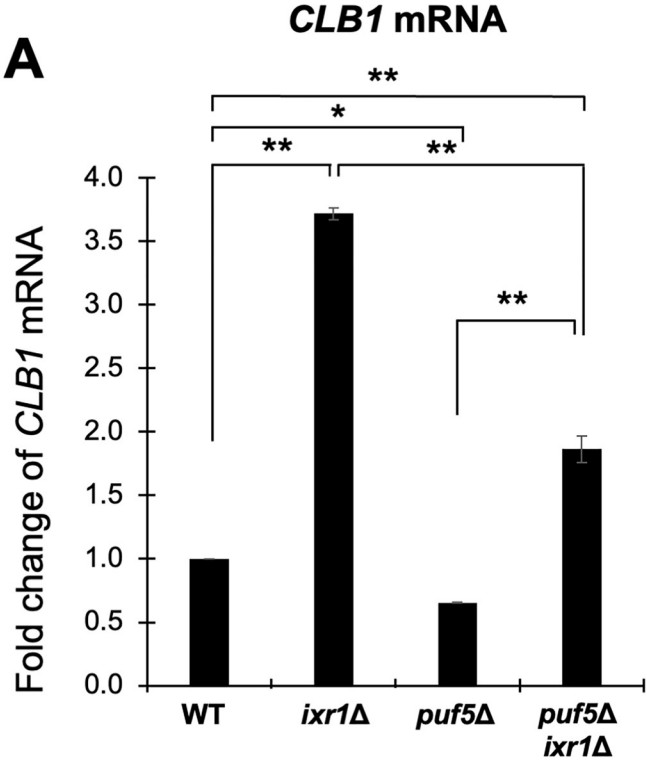

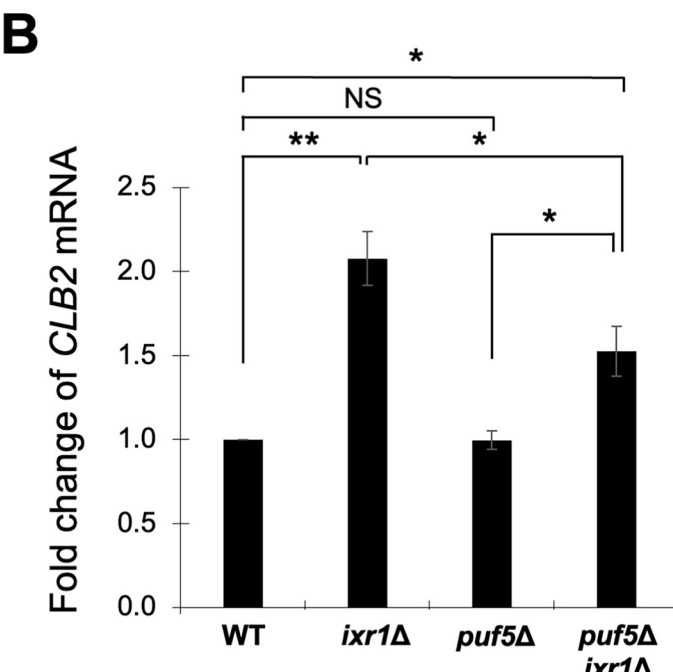

**Fig 4. Puf5 positively regulates the expression of *CLB1* and *CLB2* in an Ixr1-mediated manner.** (A, B) The mRNA levels of *CLB1* (A) and *CLB2* (B) in the wild-type strain, the *ixr1Δ* mutant, the *puf5Δ* mutant, and the *puf5Δ ixr1Δ* mutant. The cells were cultured in a YPD medium at 28˚C until the log phase. The *CLB* mRNA levels were quantified by qRT-PCR analysis, and the relative mRNA levels were calculated using the *SCR1* reference gene. The data shows the mean ± SE (n = 3) of the fold change of *CLB1* mRNA (A) and *CLB2* mRNA (B) relative to the mRNA level in the wild-type strain. *P < 0.05, **P < 0.01 as determined by Tukey's test. NS indicates no significant change.

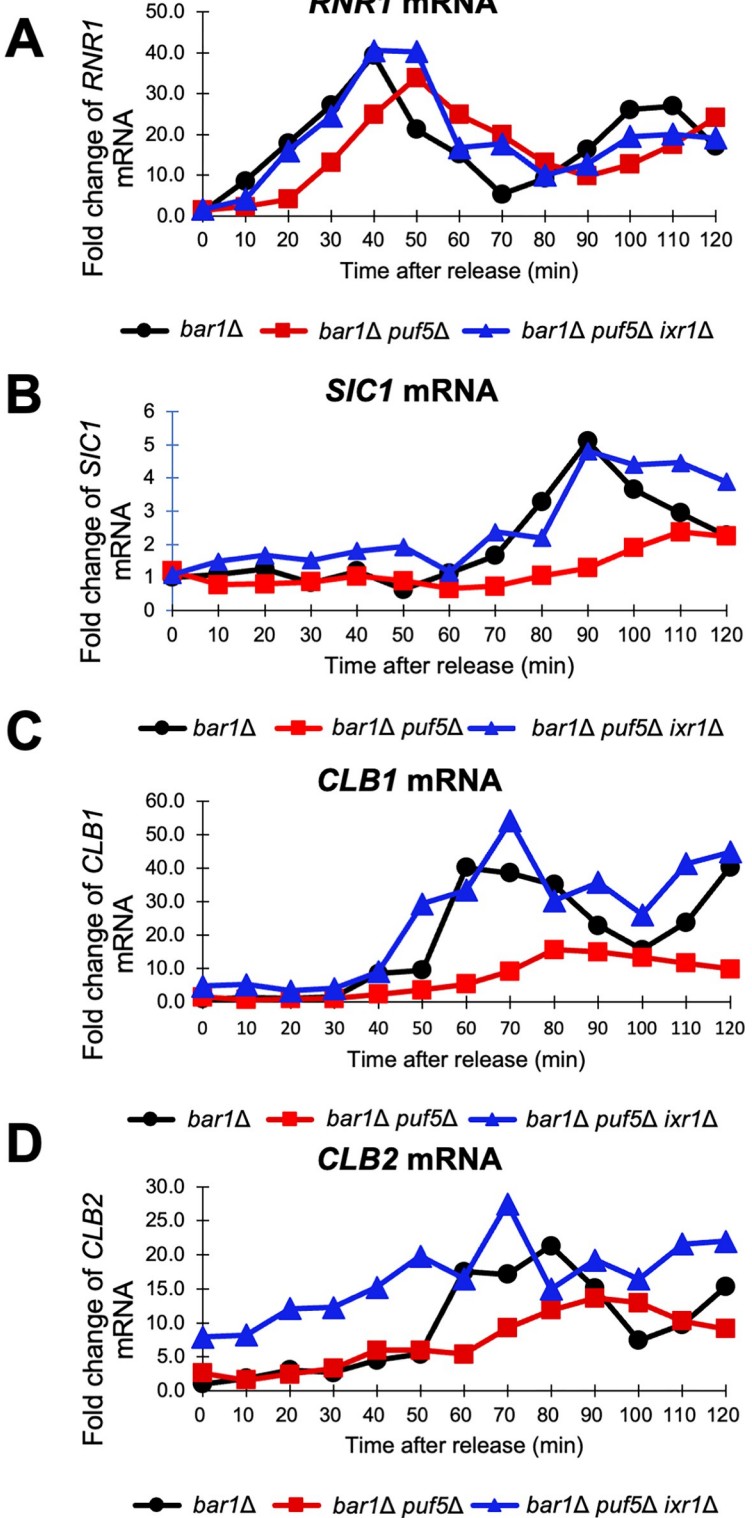

**Fig 5. The cell cycle-regulated expression of *CLB1* and *CLB2* was restored in the *puf5Δ ixr1Δ* mutant.** (A-D) The cell cycle-dependent mRNA levels of *RNR1*, *SIC1*, *CLB1*, and *CLB2* in the synchronized *bar1Δ* cell (black circle), *bar1Δ puf5Δ* mutant (red square) and *bar1Δ puf5Δ ixr1Δ* (blue triangle). The levels of an S-phase marker *RNR1* mRNA (A), a late M-phase marker *SIC1* mRNA (B), *CLB1* mRNA (C) and *CLB2* mRNA (D) were quantified by qRT-PCR analysis, and the relative mRNA levels were calculated using the *SCR1* reference gene. The vertical axis shows the fold change of

mRNA level relative to that in the *bar1Δ* 0 min sample, and the horizontal axis shows the time after release from G1-phase. These experiments were repeated (n = 2), and the representative data is presented.

*clb6Δ* triple mutant, and that the maximum expression of *CLB2* was also lower in the *puf5Δ clb1Δ clb5Δ clb6Δ* quadruple mutant (Fig 8A). Even though the *clb1Δ clb5Δ clb6Δ* triple mutant and the *puf5Δ clb1Δ clb5Δ clb6Δ* quadruple mutant lack the S-phase cyclins, the expression of the S-phase marker gene, *RNR1*, peaked at 10–20 minutes in both mutants. However, the second peak of the *RNR1* expression was only observed in the *clb1Δ clb5Δ clb6Δ* triple mutant at 110 minutes (Fig 8B), indicating that the *puf5Δ clb1Δ clb5Δ clb6Δ* quadruple mutant could not complete the first cell cycle and re-enter the second one. Indeed, the expression of the late M phase marker gene, *SIC1*, that is induced in the late M-phase, had peaked at 100 minutes in the *clb1Δ clb5Δ clb6Δ* triple mutant. However, in the *puf5Δ clb1Δ clb5Δ clb6Δ* quadruple mutant, the expression of *SIC1* was not induced during cell cycle (Fig 8C), implying that the *puf5Δ clb1Δ clb5Δ clb6Δ* quadruple mutant harbors a difficulty in the G2 to M-phase transition. To address this possibility, we observed the morphology of the two strains with nuclear staining by DAPI. The *clb1Δ clb5Δ clb6Δ* triple mutant cells showed mildly elongated shapes (S3A Fig). In this triple mutant strain, small buds were observed from 40 minutes, and the buds enlarged at 60 minutes. At 60–80 minutes, nuclei were observed near the bud neck, and nuclear divisions were seen at 100 minutes. At 120 minutes, cell divisions were completed (S3A Fig). This time course was consistent with the mRNA levels of the G2-phase-induced *CLB2* and late M-phase-induced *SIC1*. In contrast, in the *puf5Δ clb1Δ clb5Δ clb6Δ* quadruple mutant, some cells were severely elongated, and the others did just mildly. The former sometimes contained two nuclei in one cell, indicating the difficulty in the cell division in this mutant (S3B Fig). The latter contained one nucleus within each cell. As for these cells, small buds emerged from 40 minutes, and the growth of the buds was observed at 60 minutes. The nuclei were located near the bud neck at 60–80 minutes and nuclear divisions started to be observed at 80–100 minutes. However, cells could not complete cell division even at 120 minutes (S3B Fig). These results suggest that cell cycle synchronization was successfully performed in the *clb1Δ clb5Δ clb6Δ* triple mutant and the *puf5Δ clb1Δ clb5Δ clb6Δ* quadruple mutant cells, and that the *puf5Δ clb1Δ clb5Δ clb6Δ* quadruple mutant was not able to accomplish M-phase.

As previously mentioned, *CLB2* expression is downregulated in the *puf5Δ clb1Δ clb5Δ clb6Δ* quadruple mutant, and the strain shows a severe growth defect. In this case, does the decreased expression of *CLB2* cause the growth defect by itself? To clarify this question, we introduced a multi-copy *CLB2* plasmid and a multi-copy *PUF5* plasmid into the *puf5Δ clb1Δ clb5Δ clb6Δ* quadruple mutant strain. We observed that the slow growth of the *puf5Δ clb1Δ clb5Δ clb6Δ* quadruple mutant was restored by both plasmids (Fig 9). Thus, the growth retardation of the *puf5Δ clb1Δ clb5Δ clb6Δ* quadruple mutant was associated with a decrease in *CLB2* expression caused by the *puf5Δ* mutation. However, the suppression was stronger on the introduction of the multi-copy *PUF5* plasmid than the multi-copy *CLB2* plasmid. This implies the existence of other regulation targets of Puf5 on cell growth than *CLB2*. Nevertheless, our findings indicate that the appropriate expression of *CLB2* under the Puf5 control is physiologically important in the strain which lacks Clb1, Clb5, and Clb6.

## Clb2 can shoulder the function of S-phase cyclins, Clb5 and Clb6

Based on the aforementioned experiments, we found that Clb2 is essential when S-phase cyclins, Clb5 and Clb6, are lacked, and vice versa (Fig 7A). Does this imply that the G2/M-phase cyclins and S-phase cyclins complement each other's function? The *puf5Δ* mutation causes a severe growth defect in the background where Clb2 or Clb5, the major counterpart of

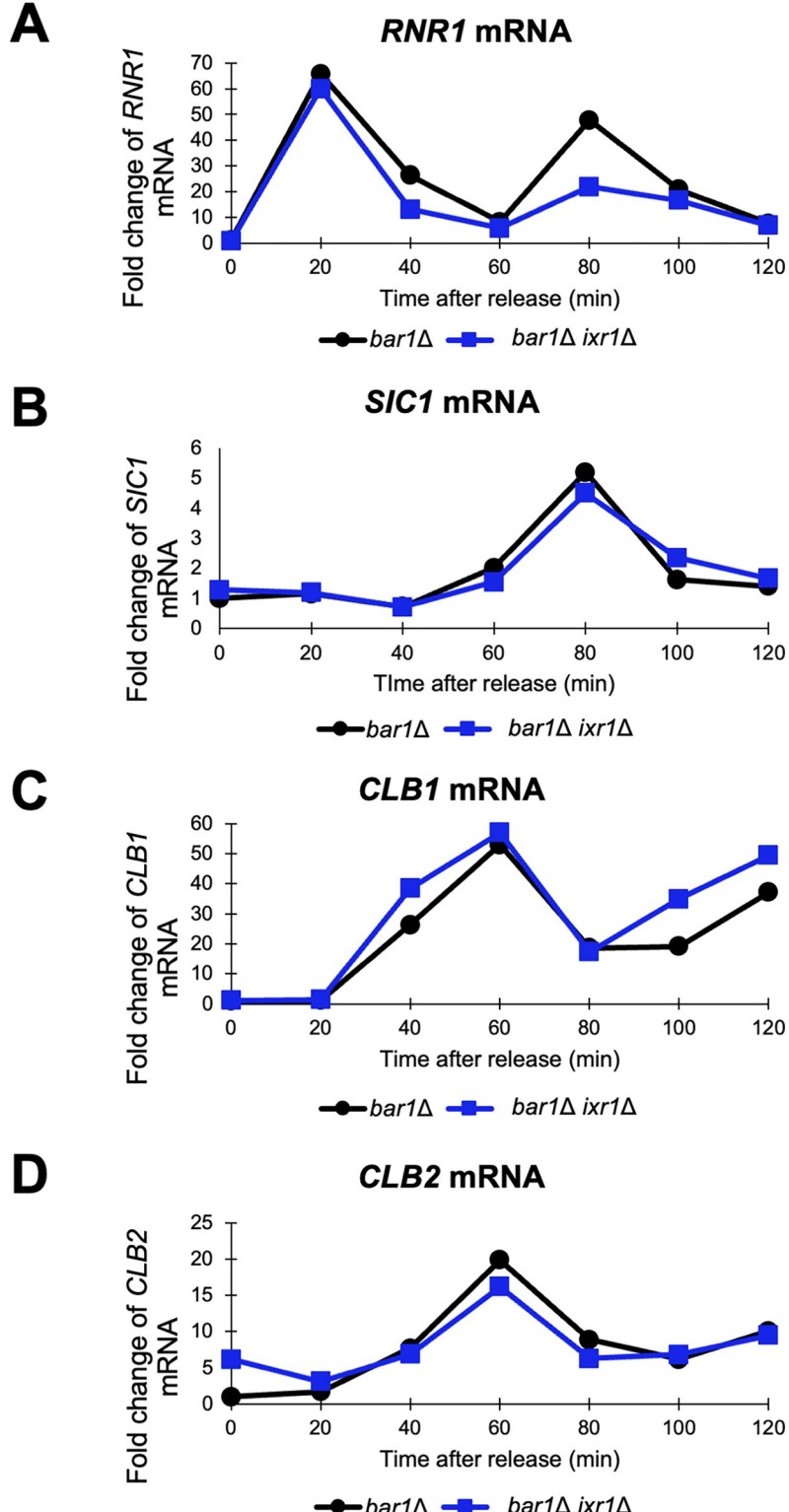

**Fig 6. The cell cycle-regulated expression of *CLB1* and *CLB2* was promoted earlier in the *ixr1Δ* mutant.** (A-D) The cell cycle-dependent mRNA levels of *RNR1*, *SIC1*, *CLB1*, and *CLB2* in the synchronized *bar1Δ* cell (black circle) and *bar1Δ ixr1Δ* mutant (blue square). The levels of an S-phase marker *RNR1* mRNA (A), a late M-phase marker *SIC1* mRNA (B), *CLB1* mRNA (C), and *CLB2* mRNA (D) were quantified by qRT-PCR analysis, and the relative mRNA levels were calculated using the *SCR1* reference gene. The vertical axis shows the fold change of mRNA level relative to

that in the *bar1Δ* 0 min sample, and the horizontal axis shows the time after release from G1-phase. These experiments were repeated (n = 2), and the representative data is presented.

the G2/M-phase cyclin pairs or S-phase cyclin pairs, is absent (Figs 1B and 7A). Therefore, we investigated the difference of the effect of each cyclin on the cell growth by making these cyclin genes overexpressed using multi-copy plasmids in the *puf5Δ clb5Δ* and the *puf5Δ clb2Δ* double mutant. The *puf5Δ clb5Δ* double mutant grew slowly even in the optimal temperature (Fig 7A), but this strain showed a remarkable growth defect at high temperature (Fig 10A). This slow growth was recovered by the multi-copy *CLB2*, *CLB5*, and *CLB6* plasmids at 35˚C (Fig 10A). At 37˚C, the *puf5Δ clb5Δ* double mutant harboring a multi-copy *CLB6* plasmid grew slightly better than cells with a vector, but the suppression by the multi-copy *CLB6* was much weaker than by the multi-copy *CLB2* and *CLB5* (Fig 10A). Therefore, the effect of the growth suppression by multi-copy *CLB2* and *CLB5* plasmids was suggested to be stronger than multi-copy *CLB6*. The multi-copy *CLB1* could not suppress the growth defect (Fig 10A). Next, we transformed the multi-copy *CLB1*, *CLB2*, *CLB5*, and *CLB6* plasmids into the *puf5Δ clb2Δ* double mutant which showed growth retardation at high temperatures. The growth of the *puf5Δ clb2Δ* double mutant was recovered at higher temperatures by the multi-copy *CLB1* and *CLB2* plasmids, but not by the multi-copy *CLB5* and *CLB6* plasmids (Fig 10B). These results suggest that Clb2, yet not Clb1, can shoulder the function of S-phase cyclins, Clb5 and Clb6, whereas Clb5 and Clb6 cannot substitute for the function of G2/M-phase cyclins, Clb2 and Clb1.

## The appropriate expression of *CLB2* was vital for the cell growth in the *dun1Δ* background

The above results suggest that Puf5 regulates cell cycle-specific expression of *CLB1* and *CLB2* via regulating the expression of Ixr1. Moreover, Clb2 is able to substitute for the S-phase cyclins, Clb5 and Clb6, and the expressional control of Clb2 by Puf5 is essential in an S-phase cyclin-deficient condition. From these, we hypothesized that the regulation of *CLB1/2* expression by Puf5 and Ixr1 is related to the differential use of Clb1/2 and Clb5/6 functions. However, no growth inhibition is observed in the *ixr1Δ* mutant, despite the increased expression of *CLB1* and *CLB2* (Figs 4A, 4B, 6C and 6D). Therefore, we further analyzed the physiological importance of Ixr1 in this B-type cyclin regulation.

The *ixr1Δ* mutation alone does not affect proliferation, but the *ixr1Δ dun1Δ* double mutant is reported to be lethal [27]. *DUN1* gene encodes a checkpoint kinase functioning downstream of Mec1-Rad53. When cell cycle checkpoint is induced, Mec1 (ATR) and Rad53 (CHEK2) are activated. This Mec1-Rad53 pathway then activates Dun1 kinase to regulate the dNTP level via upregulating the expression of *RNR* genes encoding subunits of the ribonucleotide reductase [28–30]. It is reported that in the *ixr1Δ* mutant, *RNR1* expression is decreased, which results in the low level of dNTP pool, and this regulation contributes to the lethality of the *ixr1Δ dun1Δ* double mutant [27]. However, in our data, the *RNR1* expression remains preserved in the *ixr1Δ* mutant during the first cell cycle, and the decrease was only observed in the second cell cycle (Fig 6A). Therefore, there are supposed to be other causes of the growth retardation of the *ixr1Δ dun1Δ* double mutant. We then considered and tried to verify the possibility that elevated expression of *CLB1* and *CLB2* in the *ixr1Δ* mutant was involved in the lethality of the *ixr1Δ dun1Δ* double mutant. As previously reported [27], the *ixr1Δ dun1Δ* double mutant was lethal (Fig 11A). As we expected, this lethality was also restored by the deletion of *CLB2* (Fig 11A). Nevertheless, the *clb1Δ* deletion did not recover the growth of *ixr1Δ dun1Δ* double

## A  *PUF5/puf5Δ CLB5/clb5Δ CLB6/clb6Δ*

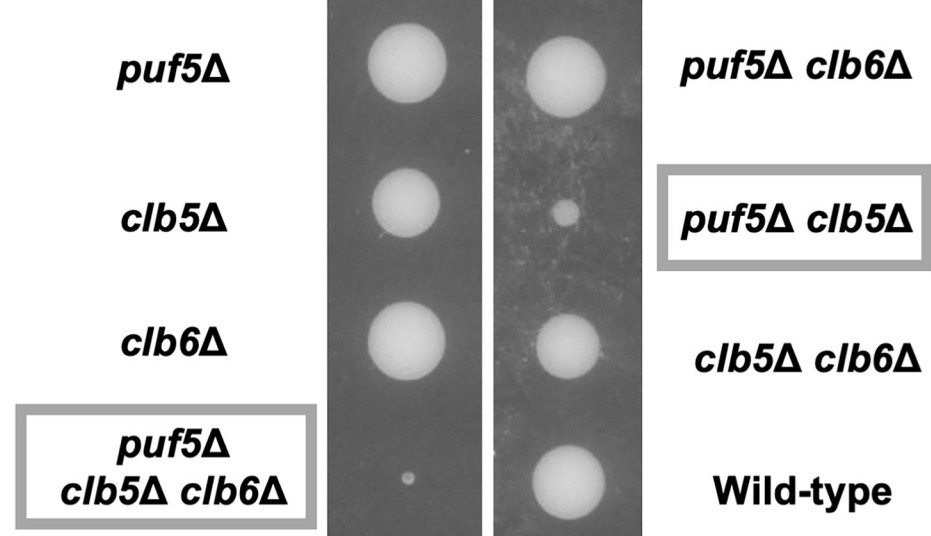

## B  *PUF5/puf5Δ CLB1/clb1Δ CLB5/clb5Δ CLB6/clb6Δ*

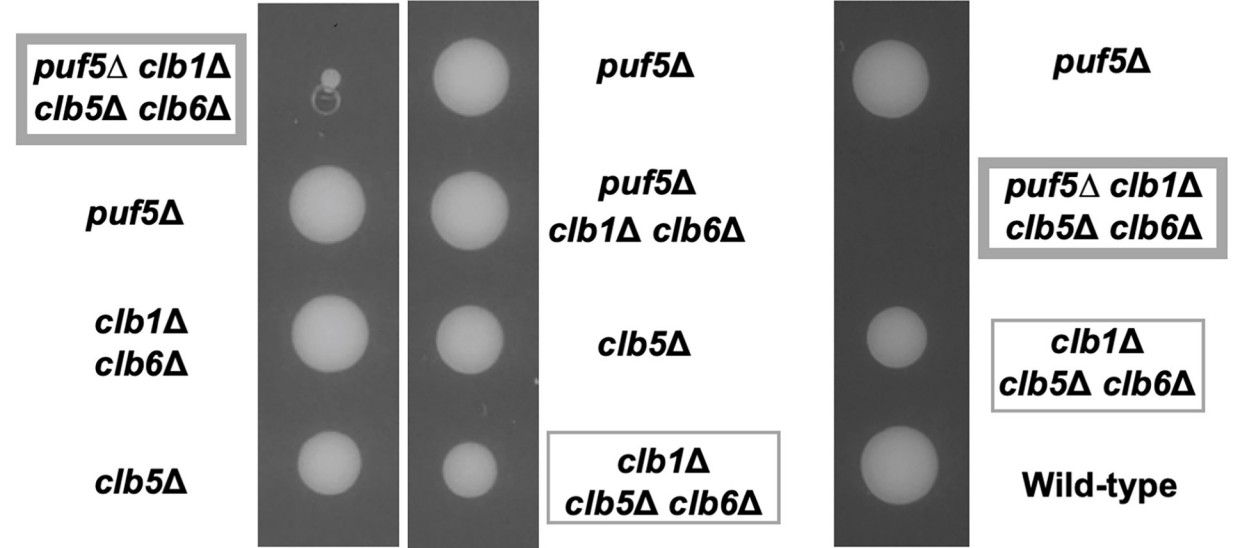

**Fig 7. The *puf5Δ* mutation caused a growth defect in an S-phase cyclin-deficient background.** (A) The tetrad analysis of the strains that are heterozygous for the alleles of *PUF5*, *CLB5*, and *CLB6*. The cells were sporulated, dissected on a YPD plate, and cultured at 30˚C for 3 days. The *puf5Δ clb5Δ* double mutant and the *puf5Δ clb5Δ clb6Δ* triple mutant were surrounded by the wide frame. (B) The tetrad analysis of the strains that are heterozygous for the alleles of *PUF5*, *CLB1*, *CLB5*, and *CLB6*. The cells were sporulated, dissected on a YPD plate, and cultured at 30˚C for 3 days. The *puf5Δ clb1Δ clb5Δ clb6Δ* quadruple mutant is emphasized in a wide frame, and the *clb1Δ clb5Δ clb6Δ* triple mutant is in the thin frame.

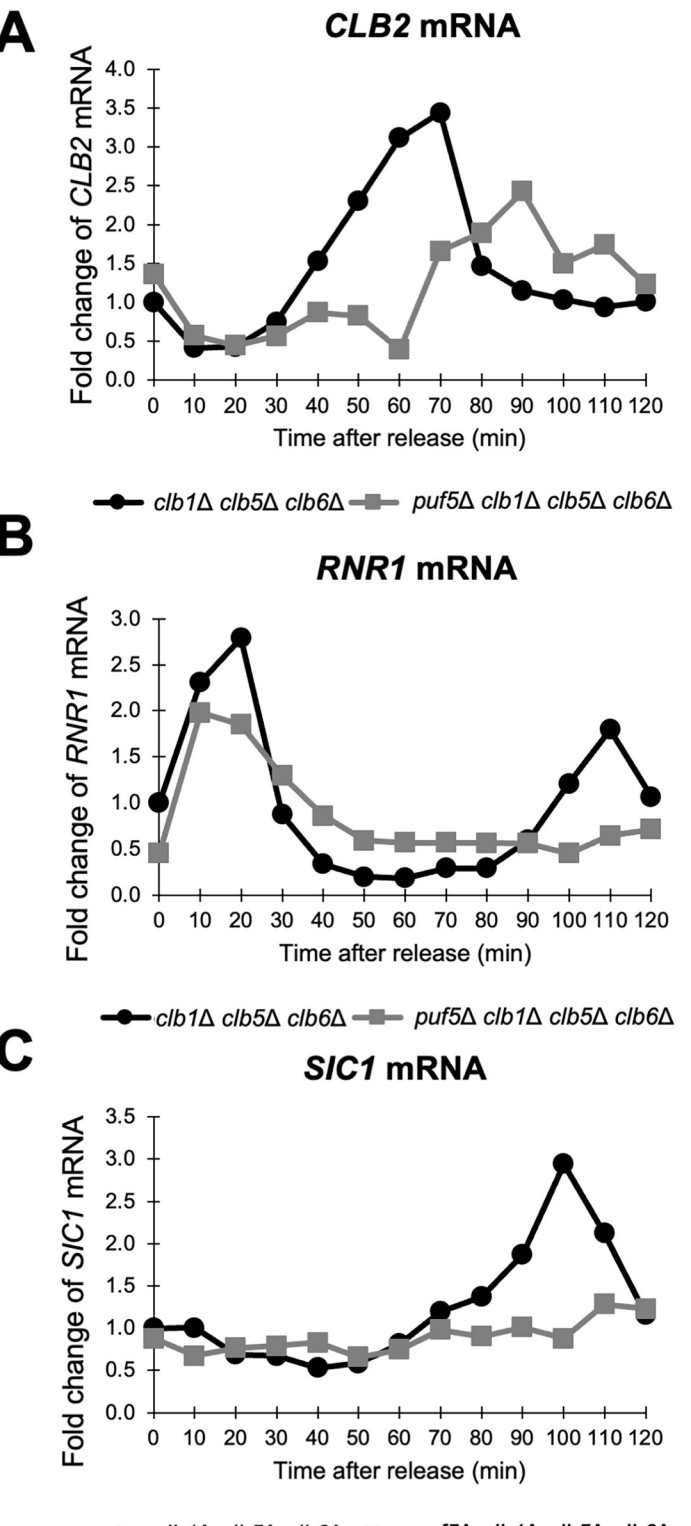

**Fig 8. The cell cycle-regulated expression of *CLB2* was diminished in the *puf5Δ clb1Δ clb5Δ clb6Δ* quadruple mutant.** (A–C) The cell cycle-dependent mRNA levels of *CLB2*, *RNR1*, and *SIC1* in the synchronized *clb1Δ clb5Δ clb6Δ* triple mutant (black circle) and the *puf5Δ clb1Δ clb5Δ clb6Δ* quadruple mutant (grey square). The levels of *CLB2* mRNA (A), S-phase marker *RNR1* mRNA (B), and a late M-phase marker *SIC1* mRNA (C) were quantified by qRT-PCR analysis, and the relative mRNA levels were calculated using the *SCR1* reference gene. The vertical axis

shows the fold change of mRNA level relative to that in the *clb1Δ clb5Δ clb6Δ* triple mutant 0 min sample, and the horizontal axis shows the time after release from G1-phase. These experiments were repeated (n = 2), and the representative data is presented.

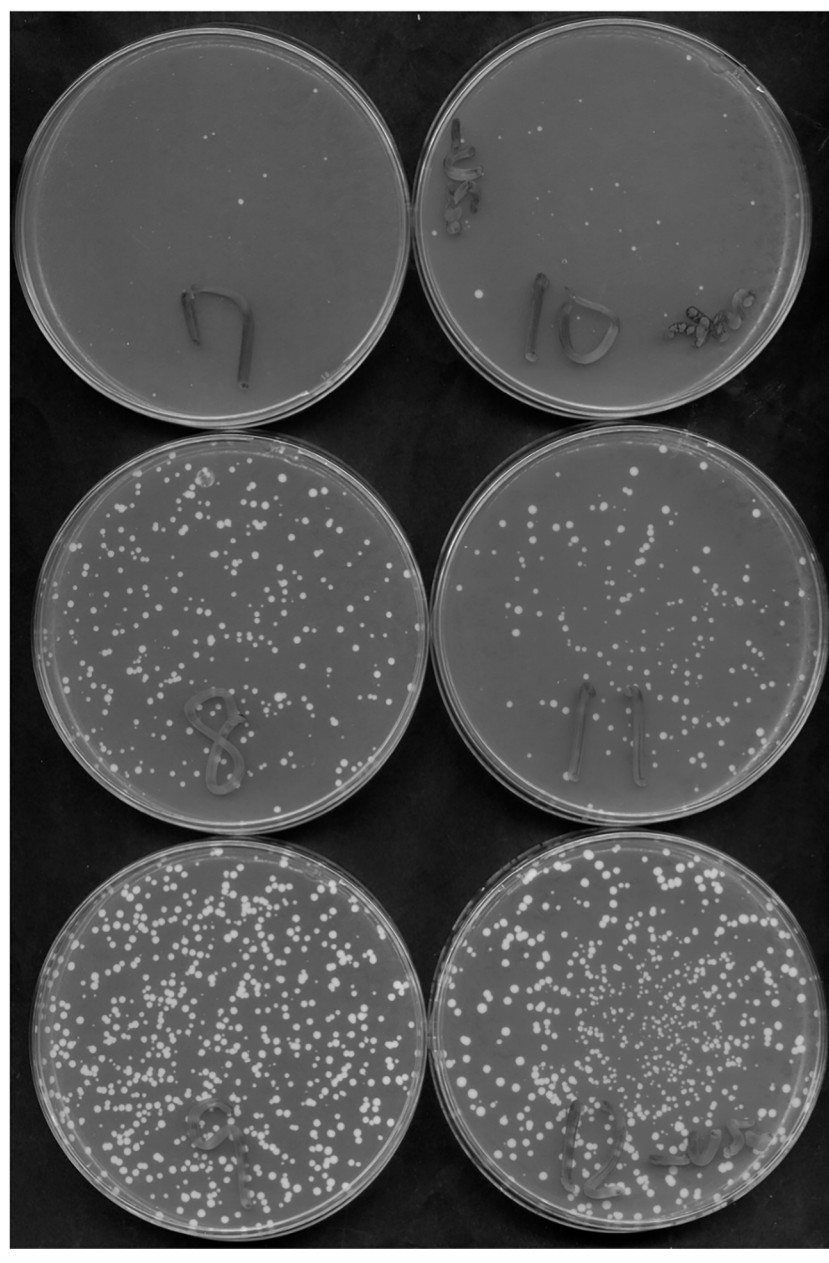

**Fig 9. *CLB2* overexpression restored the slow growth of the *puf5Δ clb1Δ clb5Δ clb6Δ* quadruple mutant.** Cell growth of the *puf5Δ clb1Δ clb5Δ clb6Δ* quadruple mutant harboring plasmids YEplac195, YEplac195-*CLB2*, or YEplac195-*PUF5*. The transformants were incubated on an SC-Ura medium at 25°C for 4 days.

### *puf5Δ clb5Δ* mutant

### *puf5Δ clb2Δ* mutant

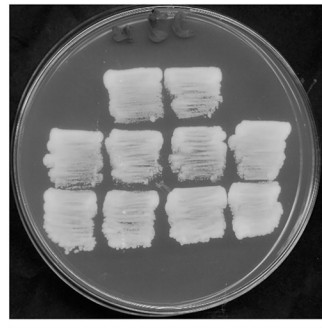
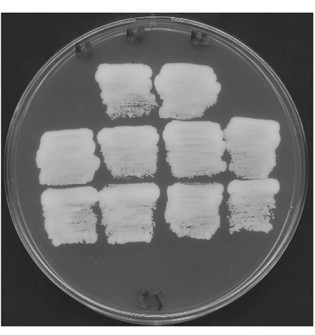
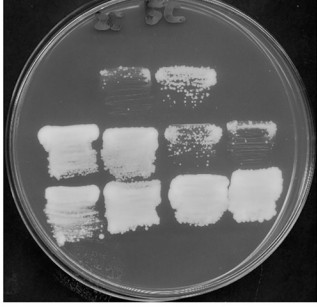
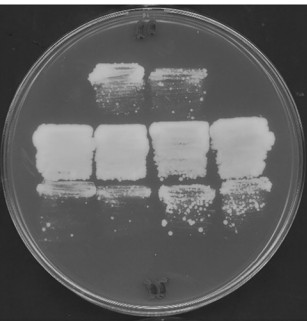
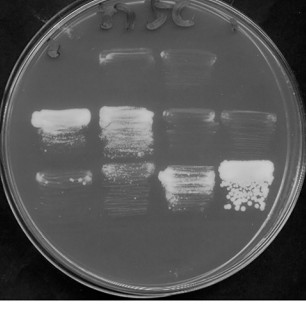
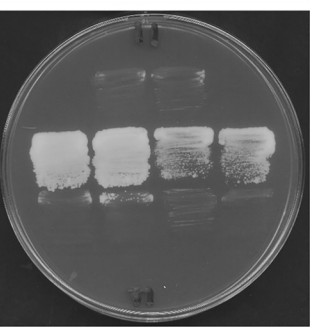

**Fig 10. *CLB2* overexpression restored the slow growth of the *puf5Δ clb5Δ* double mutant.** (A) Cell growth of the *puf5Δ clb5Δ* double mutant harboring plasmids YEplac195, YEplac195-*CLB1*, YEplac195-*CLB2*, YEplac195-*CLB5*, or YEplac195-*CLB6*. The transformants were incubated on an YPD medium at 25˚C, 35˚C, or 37˚C for 3 days. (B) Cell growth of the *puf5Δ clb2Δ* double mutant harboring plasmids YEplac195, YEplac195-*CLB1*, YEplac195-*CLB2*, YEplac195-*CLB5*, or YEplac195-*CLB6*. The transformants were incubated on an YPD medium at 25˚C, 35˚C, or 37˚C for 3 days.

mutant (S4 Fig). These results suggest that elevated *CLB2* expression, but not *CLB1*, caused by the *ixr1Δ* mutation may be associated with the lethality of the *ixr1Δ dun1Δ* double mutant. We next introduced a single copy or a multi-copy *CLB2* plasmid into the *ixr1Δ dun1Δ clb2Δ* triple mutant. Although a single-copy *CLB2* plasmid only slightly inhibited the growth of the *ixr1Δ dun1Δ clb2Δ* triple mutant, the growth repression effect of a multi-copy *CLB2* plasmid was much significant (Fig 11B). Therefore, the high dosage of *CLB2* expression seems to be harmful for the cell growth in the *dun1Δ* mutation background.

In a previous study [27], one of the causes of the lethality caused by the *ixr1Δ dun1Δ* double mutation was explained by the reduced *RNR1* expression, and the deletion of *SML1* gene encoding a ribonucleotide reductase inhibitor was reported to suppress the lethality of the *ixr1Δ dun1Δ* double mutant. Thus, we compared the suppression by the *sml1Δ* deletion with that by the *clb2Δ* deletion. We observed that the *ixr1Δ dun1Δ sml1Δ* triple mutant grew better than the *ixr1Δ dun1Δ clb2Δ* triple mutant (Fig 12A). Furthermore, the additional deletion of

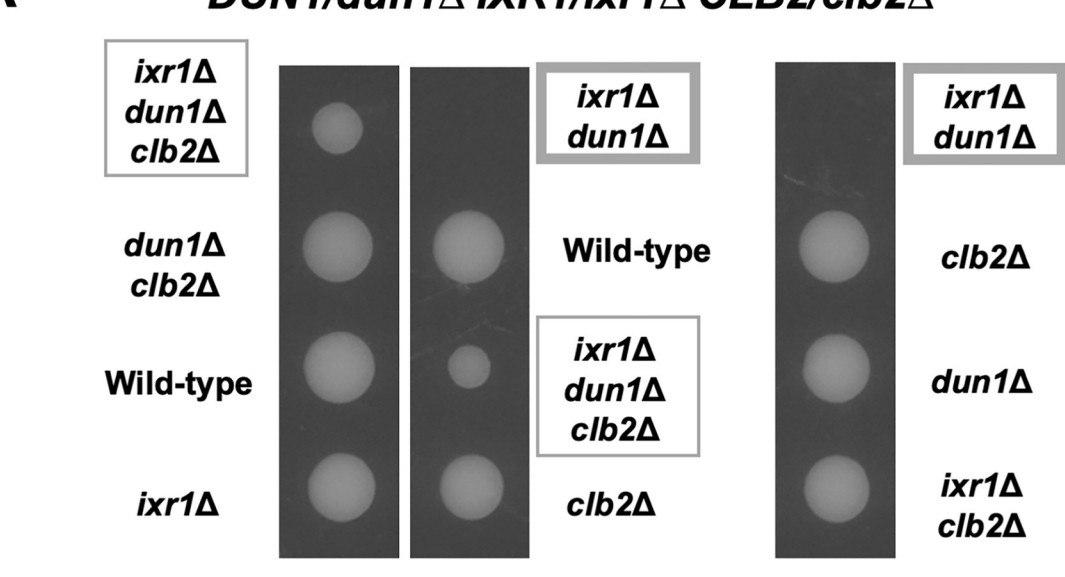

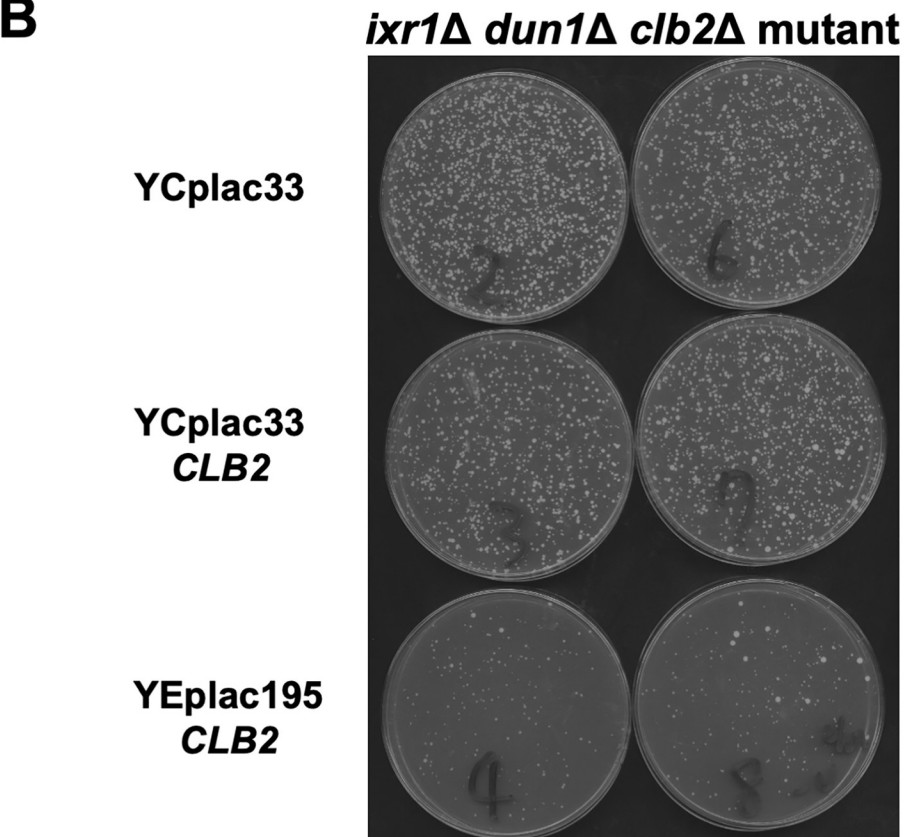

**Fig 11. Deletion of *CLB2* suppressed the lethality of the *ixr1Δ dun1Δ* double mutant.** (A) The tetrad analysis of the strains that are heterozygous for the alleles of *IXR1*, *DUN1*, and *CLB2*. The cells were sporulated, dissected on a YPD plate, and cultured at 30°C for 3 days. The *ixr1Δ dun1Δ* double mutant was emphasized with the wide frame, and the *ixr1Δ dun1Δ clb2Δ* triple mutant was in the thin frame. (B) Cell growth of the *ixr1Δ dun1Δ clb2Δ* triple mutant harboring plasmids, YCplac33, YCplac33-*CLB2*, or YEplac195-*CLB2*. The transformants were incubated on an SC-Ura medium at 25°C for 4 days.

## A   *IXR1/ixr1Δ DUN1/dun1Δ SML1/sml1Δ CLB2/clb2Δ*

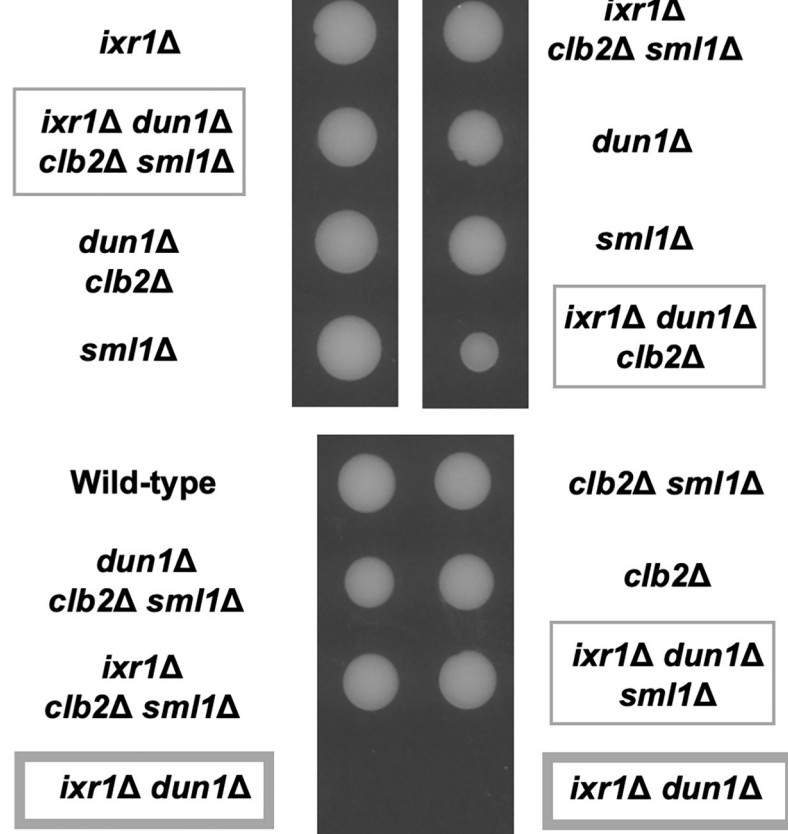

## B   *MEC1/mec1Δ SML1/sml1Δ CLB2/clb2Δ*

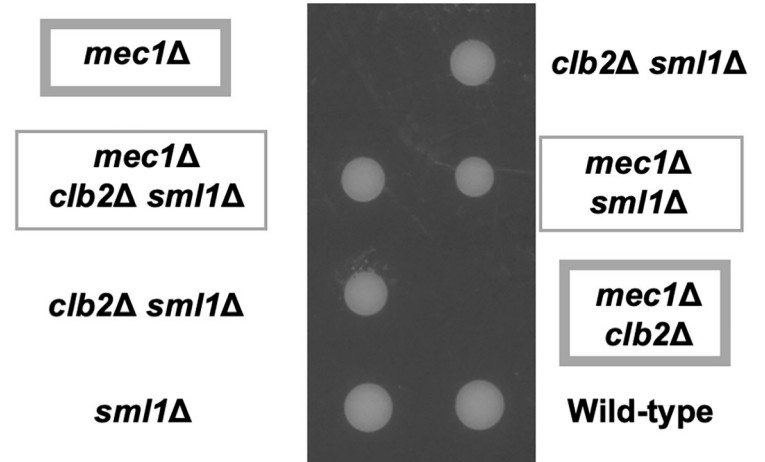

**Fig 12. The *ixr1Δ dun1Δ sml1Δ* triple mutant strain grew better than the *ixr1Δ dun1Δ clb2Δ* triple mutant strain.** (A) The tetrad analysis of the strains that are heterozygous for the alleles of *IXR1*, *DUN1*, *CLB2*, and *SML1*. The cells were sporulated, dissected on a YPD plate, and cultured at 30˚C for 3 days. The *ixr1Δ dun1Δ* double mutant was emphasized with the wide frame. (B) The tetrad analysis of the strains that are heterozygous for the alleles of *MEC1*, *CLB2*, and *SML1*. The cells were sporulated, dissected on a YPD plate, and cultured at 30˚C for 3 days. The *mec1Δ* single mutant and the *mec1Δ clb2Δ* double mutant was emphasized in a wide frame.

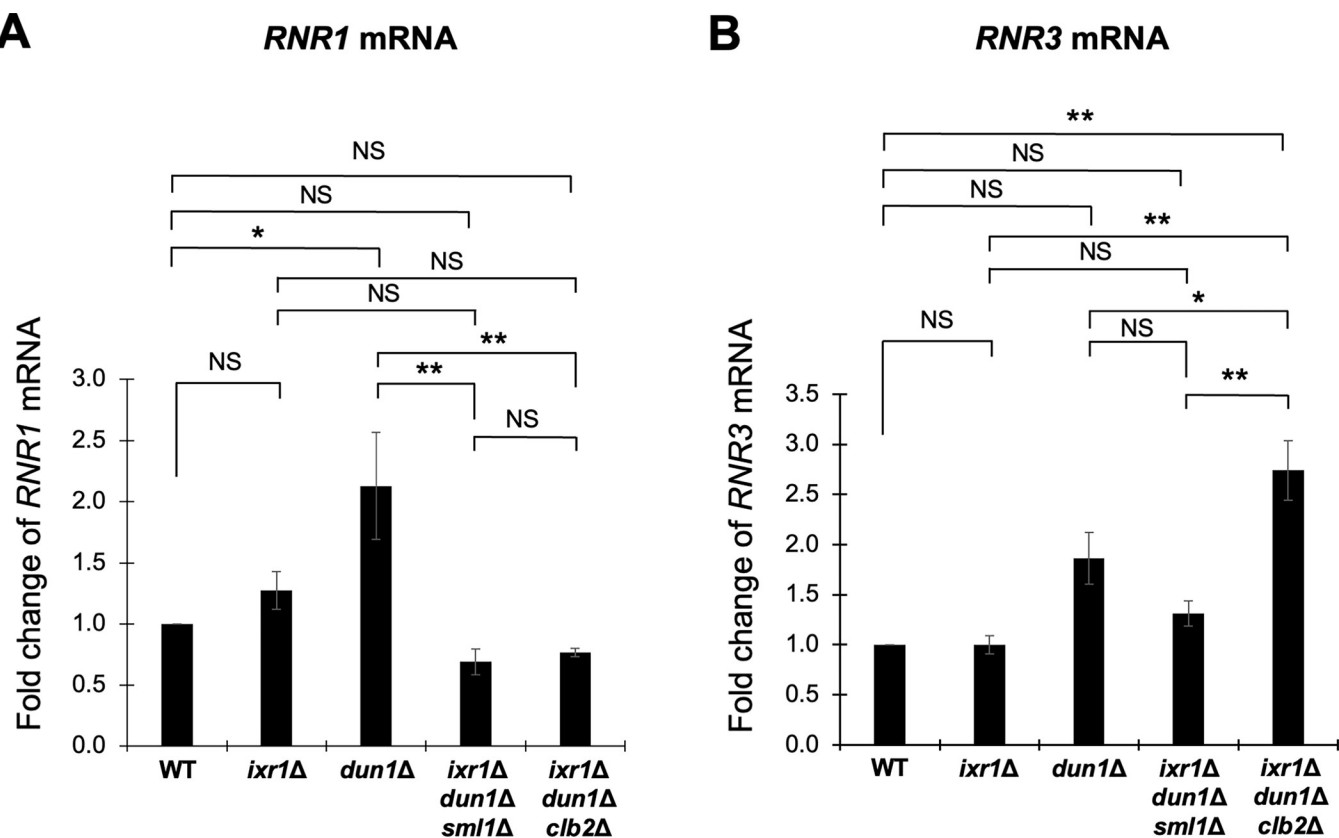

**Fig 13. The expression of *RNR3* but not *RNR1* was increased in the poorly growing *ixr1Δ dun1Δ clb2Δ* triple mutant strain.** (A, B) The mRNA levels of *RNR1* (A), and *RNR3* (B) in the wild-type strain, the *ixr1Δ* mutant, the *dun1Δ* mutant, the *ixr1Δ dun1Δ sml1Δ* triple mutant, and the *ixr1Δ dun1Δ clb2Δ* triple mutant. The cells were cultured in a YPD medium at 28°C until the log phase. The *RNR* mRNA levels were quantified by qRT-PCR analysis, and the relative mRNA levels were calculated using the *ACT1* reference gene. The data shows the mean ± SE (n = 3) of the fold change of *RNR1* (A) and *RNR3* (B) relative to the mRNA level in the wild-type strain. *P < 0.05, **P < 0.01 as determined by Tukey's test. NS indicates no significant change.

*SML1* further recovered the growth of the *ixr1Δ dun1Δ clb2Δ* triple mutant strain (Fig 12A). Thus, even though the effect of the *clb2Δ* deletion was weaker than that of the *sml1Δ* deletion, each of them independently functions in this suppression. Originally, the *sml1Δ* mutation was identified as a suppressor mutation which restored the lethality caused by *mec1Δ* deletion [29–31]. Therefore, we next examined whether the *clb2Δ* deletion also suppresses the lethality caused by *mec1Δ* deletion. Tetrad analysis showed that the *mec1Δ* single mutant was lethal, and the *mec1Δ sml1Δ* double mutant was viable as reported [31]. In contrast, the *mec1Δ clb2Δ* double mutant was lethal (Fig 12B). Thus, the effect by the *clb2Δ* deletion seemed to be different from the effect by the *sml1Δ* deletion.

To further analyze the difference between the *clb2Δ* mutation and the *sml1Δ* mutation, we examined the expression of *RNR1* and *RNR3* genes in the *ixr1Δ dun1Δ clb2Δ* triple mutant and the *ixr1Δ dun1Δ sml1Δ* triple mutant. *RNR1* expression is known to be regulated by both Ixr1 and Sml1: Ixr1 positively and Sml1 negatively regulate its expression. In contrast, the expression of *RNR3* encoding the functional partner of Rnr1 is under control of Mec1-Rad53 pathway, but not under Ixr1 or Sml1 [27]. The expression of *RNR1* was not increased in the *ixr1Δ* mutant compared to wild-type strain but was increased in the *dun1Δ* mutant (Fig 13A). As for the triple mutants of interest, *RNR1* expression was significantly decreased in both the *ixr1Δ dun1Δ clb2Δ* triple mutants and the *ixr1Δ dun1Δ sml1Δ* triple mutants compared to the *dun1Δ* mutant (Fig 13A). Regarding *RNR3*, the expression was highly upregulated in the *ixr1Δ dun1Δ*

*clb2Δ* triple mutant compared to wild-type strain and each single mutant, *ixr1Δ* mutant or *dun1Δ* mutant (Fig 13B). In contrast, in the *ixr1Δ dun1Δ sml1Δ* triple mutant, the *RNR3* level was not induced (Fig 13B). Even though *RNR1* expression levels showed a similar pattern between the *ixr1Δ dun1Δ clb2Δ* triple mutant and the *ixr1Δ dun1Δ sml1Δ* triple mutant, the converse expression of *RNR3*, a gene controlled by Mec1-Rad53 checkpoint pathway, implies the difference in the intracellular condition of these mutants.

## The *dun1Δ* mutation shows a synthetic growth defect with the *clb5Δ clb6Δ* double mutation

As mentioned in the former section, we hypothesize that Puf5 and Ixr1 play an important role for the proper utilization of G2/M-phase cyclins and S-phase cyclins. The increased level of *CLB2* encoding a major G2/M cyclin caused by the *ixr1Δ* mutation is found to be toxic in the *dun1Δ* mutation background. Then, does this effect extend to the S-phase cyclins? Previously, genetical interactions among *DUN1* and S-phase cyclin genes, *CLB5* and *CLB6*, have been reported [32]. Therefore, we next examined these genetic interactions. Tetrad analysis revealed that the *dun1Δ clb5Δ clb6Δ* triple mutant had a poor growth (Fig 14A). Contrarily, the *dun1Δ clb5Δ* double mutant showed only a moderate growth retardation, and the *dun1Δ clb6Δ* double mutant grew as well as wild-type strain (Fig 14A). From these results, the expression of *CLB5* and *CLB6*, in contrast to *CLB2*, seem to have a positive effect on cell growth under the *dun1Δ* mutation background. Next, we investigated how *CLB2* expression affected the growth of the *dun1Δ clb5Δ clb6Δ* triple mutant by additionally deleting *PUF5* gene. As shown in Fig 7A, the *puf5Δ clb5Δ clb6Δ* triple mutant showed a severe growth defect, and this defect was more remarkable than in the *dun1Δ clb5Δ clb6Δ* triple mutant (Fig 14B). The *puf5Δ dun1Δ* double mutant grew as well as wild-type strain, but the *puf5Δ* mutation significantly accelerated the growth retardation of the *dun1Δ clb5Δ* double mutant and the *dun1Δ clb5Δ clb6Δ* triple mutant (Fig14B). Although the excessive expression of *CLB2* worsens the growth of *dun1Δ* mutant, proper *CLB2* expression seems to be needed to maintain cell growth in the absence of S-phase cyclins. To further clarify the difference among the effects of *CLB5/CLB6* and *CLB2* on cell growth in the *dun1Δ* background, we next examined the expression of *RNR1* and *RNR3* genes in the *dun1Δ clb5Δ clb6Δ* triple mutant, together with the *puf5Δ clb5Δ clb6Δ* triple mutant. The expression of *RNR1* did not change significantly in the *dun1Δ clb5Δ clb6Δ* triple mutant compared to the *dun1Δ* mutant (S5 Fig), while the expression of *RNR3* was significantly increased about 15 times in the *dun1Δ clb5Δ clb6Δ* triple mutant and 8 times in the *dun1Δ clb5Δ* double mutant compared to wild-type strain (Fig 15). The increased expression of *RNR3* were also significant compared to the *dun1Δ* mutant (Fig 15). Similarly, the expression of *RNR1* did not change significantly between the *puf5Δ clb5Δ clb6Δ* triple mutant and other single or double mutants (S5 Fig). In contrast, *RNR3* mRNA level was significantly higher about 2.5 times in the *puf5Δ clb5Δ* double mutant and 3 times in the *puf5Δ clb5Δ clb6Δ* triple mutant compared to wild-type strain (Fig 16). Together with the shown data in Fig 13B, the expression of *RNR3* regulated by the Mec1-Rad53 checkpoint pathway is highly induced when G2/M-phase cyclins or S-phase cyclins are absent, and this induction is particularly remarkable in the *dun1Δ* mutation conditions.

## Discussion

We previously showed that Puf5 positively regulates *CLB1* expression via the post-transcriptional regulation of *IXR1* mRNA. This *IXR1* mRNA encodes a repressor protein Ixr1 which appears to function as a negative regulator of *CLB1* expression [25]. In that paper, in an asynchronous culture, the expression of *CLB2* encoding a redundant cyclin of Clb1 is not decreased

## A  *DUN1/dun1Δ CLB5/clb5Δ CLB6/clb6Δ*

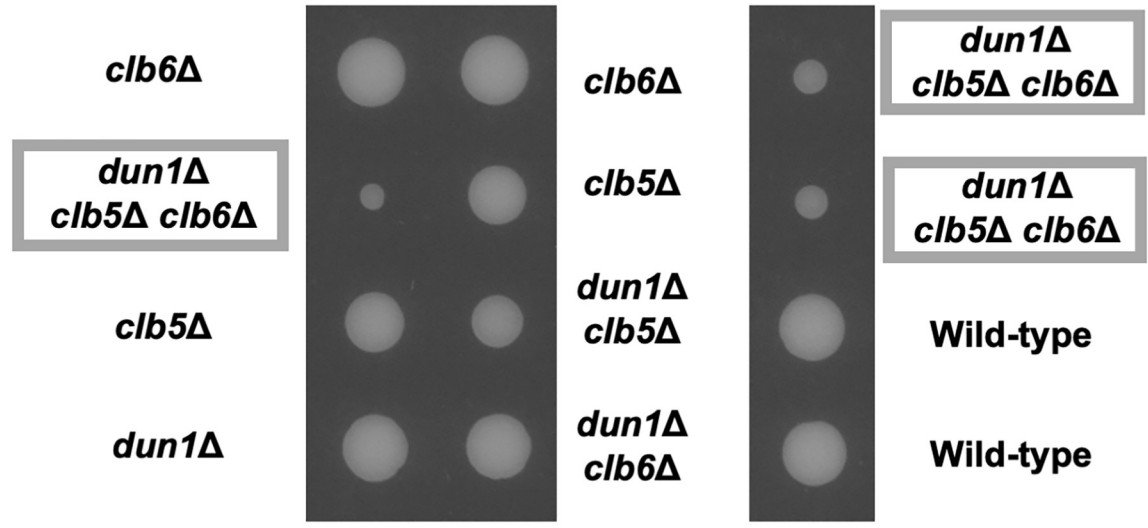

## B  *PUF5/puf5Δ DUN1/dun1Δ CLB5/clb5Δ CLB6/clb6Δ*

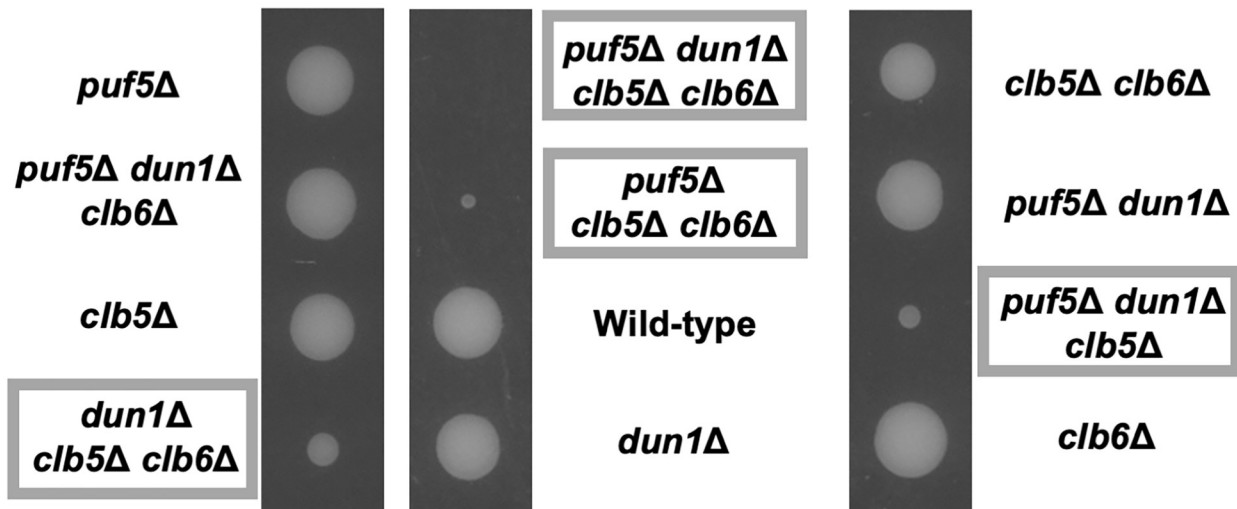

**Fig 14. The *dun1Δ clb5Δ clb6Δ* triple mutant showed poor growth.** (A) The tetrad analysis of the strains that are heterozygous for the alleles of *DUN1*, *CLB5*, and *CLB6*. The cells were sporulated, dissected on a YPD plate, and cultured at 30°C for 3 days. The *dun1Δ clb5Δ clb6Δ* triple mutant was surrounded by the wide frame. (B) The tetrad analysis of the strains that are heterozygous for the alleles of *PUF5*, *DUN1*, *CLB5*, and *CLB6*. The cells were sporulated, dissected on a YPD plate, and cultured at 30°C for 3 days.

# *RNR3* mRNA

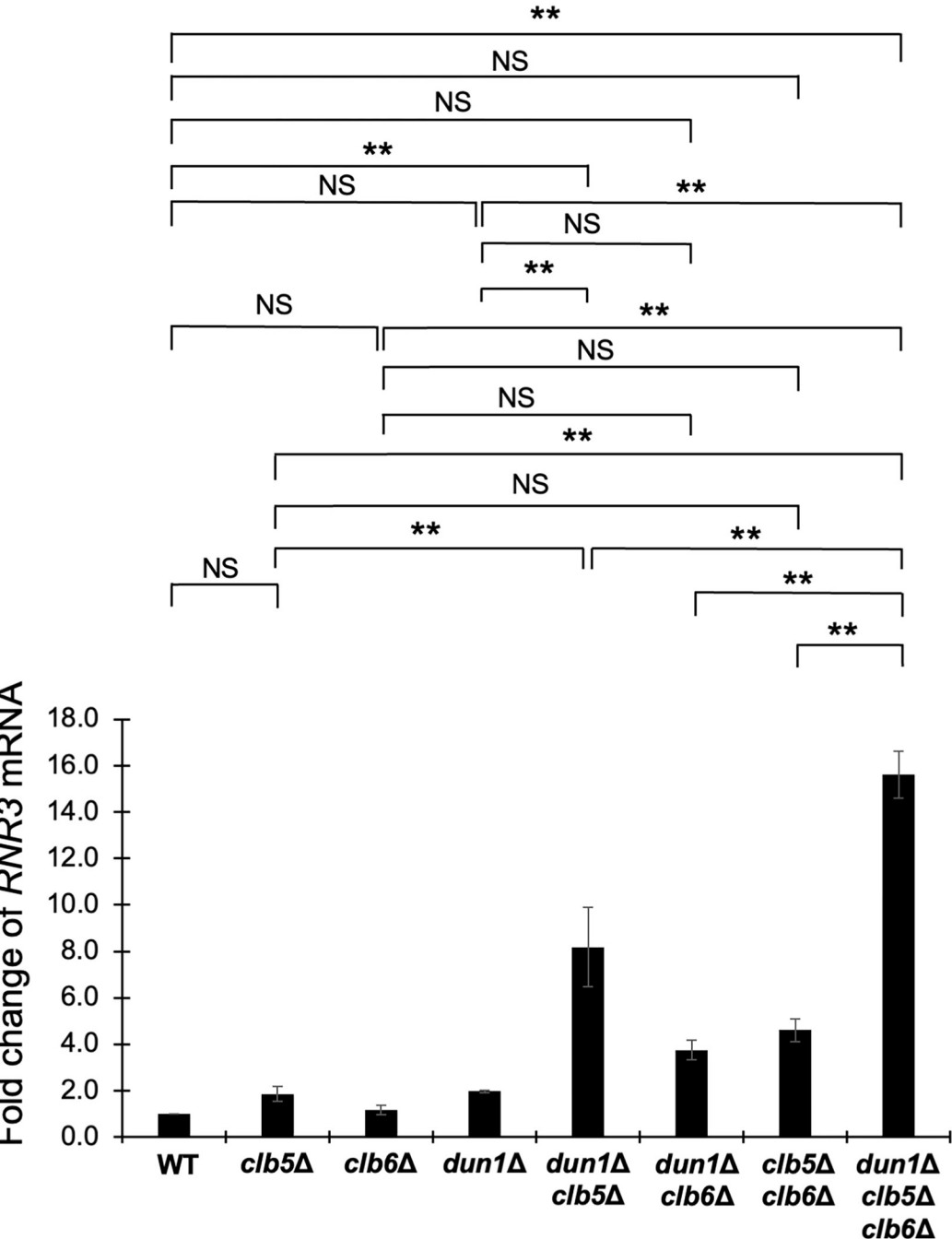

**Fig 15. The expression of *RNR3* was increased in the *dun1Δ clb5Δ clb6Δ* triple mutant.** The mRNA levels of *RNR3* in the wild-type strain, the *clb5Δ* mutant, the *clb6Δ* mutant, the *dun1Δ* mutant, the *dun1Δ clb5Δ* mutant, the *dun1Δ clb6Δ* mutant, the *clb5Δ clb6Δ* mutant, and the *dun1Δ clb5Δ clb6Δ* mutant. The cells were cultured in a YPD medium at 28°C until the log phase. The *RNR3* mRNA levels were quantified by qRT-PCR analysis, and the relative mRNA levels were calculated using the *ACT1* reference gene. The data shows the mean ± SE (n = 3) of the fold change of *RNR3* relative to the mRNA level in the wild-type strain. **P < 0.01 as determined by Tukey's test. NS indicates no significant change.

## *RNR3* mRNA

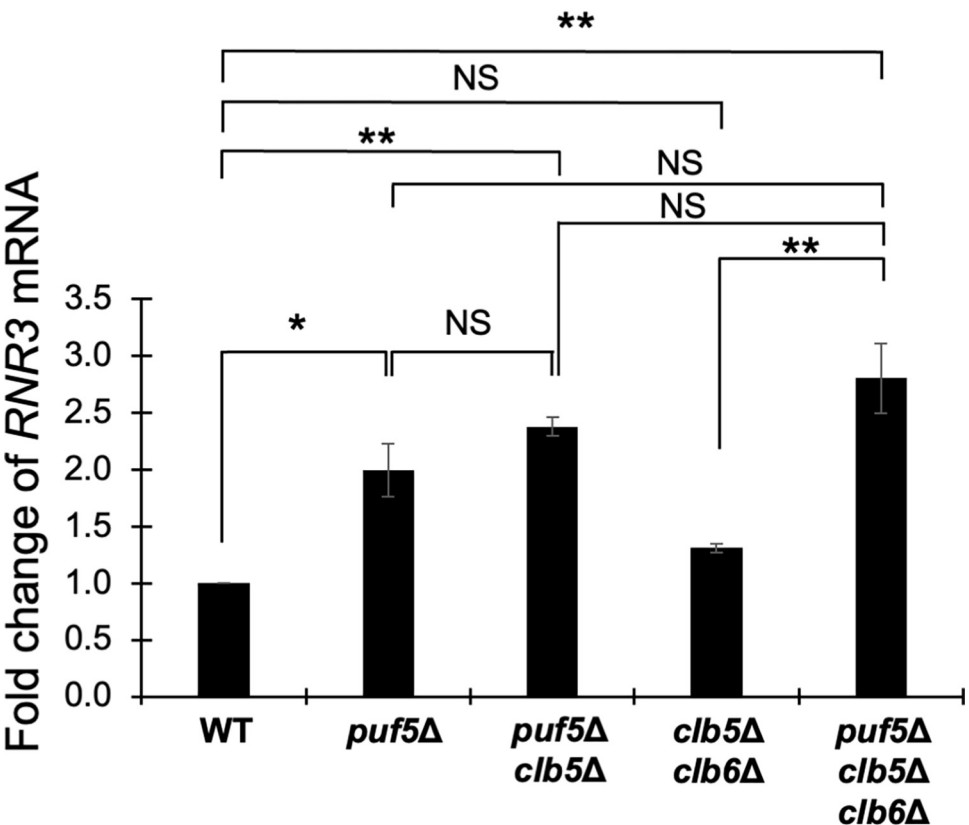

**Fig 16. The expression of *RNR3* was increased in the *puf5Δ clb5Δ clb6Δ* triple mutant strain.** The mRNA levels of *RNR3* in the wild-type strain, the *puf5Δ* mutant, the *puf5Δ clb5Δ* mutant, the *clb5Δ clb6Δ* mutant, and the *puf5Δ clb5Δ clb6Δ* mutant. The cells were cultured in a YPD medium at 28˚C until the log phase. The *RNR3* mRNA levels were quantified by qRT-PCR analysis, and the relative mRNA levels were calculated using the *ACT1* reference gene. The data shows the mean ± SE (n = 3) of the fold change of *RNR3* relative to the mRNA level in the wild-type strain. *P < 0.05, **P < 0.01 as determined by Tukey's test. NS indicates no significant change.

in the *puf5Δ* mutant. However, regarding that *CLB1* and *CLB2* are under the same expressional control machinery in mitotic cell proliferation, it would be reasonable that Puf5 also regulates the expression of *CLB2* in addition to *CLB1*. Thus, in this study, we re-examined whether Puf5-Ixr1 is also involved in the control of the expression of *CLB2*. As a result, we observed a decrease in cell cycle-specific expression of *CLB2* mRNA and protein in the *puf5Δ* mutant in synchronized cultures (Figs 2D and 3B). The degree of the decrease in *CLB2* expression, however, seemed to be milder than that in *CLB1* expression in both mRNA and protein levels (Figs 2D and 3B). This difference also affects the physiological importance of these two regulations. In detail, the decreased expression of *CLB1* by the *puf5Δ* mutation caused a severe growth defect in the *clb2Δ* mutation background, whereas the *puf5Δ clb1Δ* double mutant grew like wild-type strain (Fig 1B). These results imply two possibilities: one is that the remaining *CLB2* expression, albeit at the decreased level, in the *puf5Δ* mutant maintained cell proliferation, the

other is that the other B-type cyclins, such as Clb5 and Clb6, compensate for Clb1 and Clb2. Since our results indicated that Clb5 and Clb6 were not able to substitute for the G2/M-phase cyclins (Fig 10B), the former possibility is more reasonable for explaining the growth of the *puf5Δ clb1Δ* double mutant.

Seeking the physiological importance of the regulation of *CLB2* expression by Puf5, we uncovered that the decrease in the *CLB2* expression causes a growth retardation in the absence of S-phase cyclins. The *puf5Δ clb1Δ clb5Δ clb6Δ* quadruple mutant exhibited a severe growth defect or sometimes lethality (Fig 7B), and this remarkably poor growth was caused by the decreased *CLB2* expression by the *puf5Δ* mutation. However, when we introduced a multi-copy *CLB2* or a multicopy *PUF5* plasmid into the *puf5Δ clb1Δ clb5Δ clb6Δ* quadruple mutant, the suppression effect of the multi-copy *CLB2* plasmid was milder than that of multicopy *PUF5* plasmid (Fig 9). Since Puf5 binds to more than 1,000 mRNAs [24] and regulates diverse phenomena, Puf5 is supposed to regulate more targets than specified so far. Therefore, this difference in the suppression effect is implied to originate from other targets of Puf5. Nevertheless, Puf5 does positively regulate *CLB2* expression of physiological importance in the S-phase cyclin-deficient condition. Considering that Clb2, but not Clb1, could compensate for the loss of S-phase cyclin Clb5 (Fig 10A), Puf5 controls the *CLB2* expression at a proper level to ensure the functional redundancy of two major B-type cyclins, Clb2 and Clb5. The B-type cyclins in *S. cerevisiae* were supposed to evolve from common B-type cyclin like ancestors and diverge into six types during evolution [33]. The ancestors are believed to harbor both replicational and mitotic activity. Therefore, the functional redundancy between Clb2 and Clb5 is possibly an evolutionary trace. Actually, it has been reported that in the *clb5Δ clb6Δ* double mutant, the entry into the S-phase was delayed severely, but origin firing was performed normally, suggesting that there is a functional overlap between Clb1-4 and Clb5/Clb6 [34]. However, Clb2 under the control of endogenous *CLB5* promoter was not able to suppress the origin firing defect in the *clb5Δ clb6Δ* double mutant and had only a partial effect on the recovering the entry into the S-phase [35]. Considering our data that the multi-copy *CLB2* plasmid suppressed the growth defect of the *puf5Δ clb5Δ* double mutant (Fig 10A), two possibilities are suggested: one is that the high dosage of Clb2 is essential to shoulder the Clb5 function, and the other is that only a partial functional redundancy is sufficient for cell growth. We regard the former possibility more plausible, since the expression level of Clb2 is critical for maintaining cell growth in the *dun1Δ* mutation background.

In this study, we presented that Puf5 regulates *CLB2* expression in an Ixr1-dependent manner. Then, what is the significance of Ixr1 regulating the expression of *CLB1* and *CLB2*? Previously it had been reported that the *ixr1Δ* mutation causes the decreased *RNR1* expression level and results in the lethality in the *dun1Δ* mutation background, one of the checkpoint kinase-deficient condition [27]. Here, we found that the *clb2Δ* mutation, but not the *clb1Δ* mutation, suppressed the lethality of the *ixr1Δ dun1Δ* double mutant (Figs 11A and S4). The deletion of *SML1*, a negative regulator of *RNR1*, also restored the growth defect of the *ixr1Δ dun1Δ* double mutant. However, the *sml1Δ* mutation could additionally recover the growth of the *ixr1Δ dun1Δ clb2Δ* triple mutant (Fig 12A), and the suppression effects of the *mec1Δ* mutant between the *sml1Δ* mutation and the *clb2Δ* mutation were far different: the *sml1Δ* mutation could, but the *clb2Δ* mutation could not restore the lethality (Fig 12B). From these results, it is hypothesized that the growth suppression by the *clb2Δ* mutation is independently performed of Sml1. Moreover, the growth of the *ixr1Δ dun1Δ clb2Δ* triple mutant was inhibited by the multi-copy *CLB2* plasmid, whereas the inhibitory effect of the single-copy *CLB2* plasmid was not sufficient as that of the multi-copy *CLB2* plasmid (Fig 11B). Altogether, the *ixr1Δ dun1Δ* double mutant or *CLB2*-overexpressed *ixr1Δ dun1Δ clb2Δ* triple mutant showed growth defect or lethality, and the *ixr1Δ dun1Δ clb2Δ* triple mutant or single-copy *CLB2* expressed *ixr1Δ dun1Δ clb2Δ*

triple mutant was viable (Fig 11A and 11B). From these results, we suppose that the high dosage of Clb2 is critical to the *dun1Δ* mutant. In other words, it is crucial for cell proliferation to maintain the proper level of *CLB2* expression under the control by Ixr1 in the *dun1Δ* mutant. Regarding that *CLB1* expression does not have physiological importance in the *dun1Δ* mutation background (S4 Fig), the functions of Ixr1 on *CLB1* or *CLB2* seem to be different. Ixr1 regulates *CLB1* expression specifically to G2/M-phase, while regulating *CLB2* expression at the global level rather than in a cell cycle-specific way (Figs 5D and 6D). It is possible that this global effect of Ixr1 on *CLB2* expression ensures the proper level of Clb2 and contributes to the cell viability under the DNA-damage uninducible *dun1Δ* mutation background.

In contrast to *CLB2*, the expression of *CLB5* and *CLB6* positively affected the growth of the *dun1Δ* mutant: the *dun1Δ clb5Δ clb6Δ* triple mutant strain grew slowly (Fig 14A). Moreover, the *puf5Δ dun1Δ clb5Δ clb6Δ* quadruple mutant strain was also lethal (Fig 14B). Therefore, in addition to *CLB2*, the adequate expression of B-type cyclins seem to be necessary for the *dun1Δ* mutant to survive. Thus, what does happen when the expression of B-type cyclins were lost in the *dun1Δ* mutant? From the mRNA levels of *RNR3*, we hypothesize that DNA damage response is highly induced in the condition. The expression of *RNR3*, a gene encoding subunits of the ribonucleotide reductase, is generally induced by Mec1-Rad53-Dun1 checkpoint pathway and also directly induced Mec1-Rad53 in a Dun1-independent manner [28,29]. Since its expression is quite low under the non-stressed condition, the induction of *RNR3* expression is usually used as a marker for DNA damage [36]. As a result, *RNR3* expression was mildly induced in the *dun1Δ* mutant and highly induced in the *ixr1Δ dun1Δ clb2Δ* triple mutant and the *dun1Δ clb5Δ clb6Δ* triple mutant (Figs 13B and 15). This induction compared to the wildtype strain was more prominent in the *dun1Δ clb5Δ clb6Δ* triple mutant than in the *ixr1Δ dun1Δ clb2Δ* triple mutant (Figs 13B and 15). These results imply that Mec1-Rad53 dependent DNA damage response is activated when B-type cyclins are absent in the *dun1Δ* mutation background. Previously, it was reported that DNA damage activates the checkpoint pathway more strongly in the S phase than in other cell cycle phases [37]. Thus, the abnormality of DNA damage induction in the *dun1Δ* mutant is possibly highly accelerated by the absence of S-phase cyclins. In addition, *RNR3* expression was significantly increased in the *puf5Δ* mutant, the *puf5Δ clb5Δ* double mutant, and the *puf5Δ clb5Δ clb6Δ* triple mutant, while *RNR3* expression was not induced in the *clb5Δ clb6Δ* double mutant (Fig 16). Therefore, we assume that the deficiency of G2/M-phase cyclins causes DNA damage responses by themselves, and that this response is spurred by the additional deletion of *CLB5*/*CLB6*. This induced DNA damage responses possibly contributes to the growth retardation of the *puf5Δ clb5Δ* double mutant and the *puf5Δ clb5Δ clb6Δ* triple mutant. In this study, we analyzed the regulation mechanism of a B-type cyclin gene *CLB2* by Puf5 and Ixr1. We speculate that Puf5 and Ixr1 finetune expression of *CLB2* and contribute to the maintenance of the sufficient function of cyclins in the G2/M-phase and S-phase. Furthermore, Ixr1-mediated regulation maintains adequate expression levels of *CLB2* and cell proliferation under DNA damage conditions. From our data, the deficiency of G2/M-phase cyclins is suggested to cause DNA damage by itself, so the finetuning of *CLB2* expression is assumed to be a vital process for cells to survival.

## Materials and methods

### Strains and media

The *Saccharomyces cerevisiae* strain W303 was used as the background yeast strain for the study. *Escherichia coli*, DH5α strain, was used to manipulate the DNA. The genetic manipulation of yeast strains was performed using standard procedures as previously described [38]. All W303-derived strains used in this study are described in detail in S1 Table. For standard

culture, *Saccharomyces cerevisiae* was grown in YPD medium (2% Glucose, 2% bactopeptone, and 1% yeast extract) and synthetic complete medium (SC) [38]. To culture yeast strains that require alleviation of osmolarity stress, 10% sorbitol was supplemented to the media. SC medium lacking amino acids (e.g., SC-Ura medium i.e., SC medium lacking uracil) were used to select transformants.

## Plasmids

The plasmids used in this study are described in S2 Table. For the construction of YEplac195-*PUF5* plasmid, the fragment containing the *PUF5* gene together with upstream and downstream regions was amplified by PCR of genomic DNA. The fragment was inserted between *Sal*I and *Eco*RI sites of YEplac195 plasmid. YEplac195-*CLB1*, YEplac195-*CLB2*, YEplac195-*CLB5*, and YEplac195-*CLB6* plasmids were constructed following similar procedure. For the construction of YCplac33-*CLB2* plasmid, the fragment containing the *CLB2* gene together with upstream and downstream regions was amplified by PCR and inserted between *Sal*I and *Eco*RI sites of YCplac33 plasmid.

The pCgLEU2, pCgHIS3, and pCgTRP1 plasmids, which were pUC19 carrying the *Candida glabrata LEU2*, *HIS3*, and *TRP1* genes, respectively, were used to delete genes [39]. The pKl-*URA3* plasmid, pUC19 carrying the *Kluyveromyces lactis URA3* were also used for gene deletion.

## Gene deletion

Deletions of *PUF5*, *IXR1*, *CLB1*, *CLB2*, *CLB3*, *CLB4*, *CLB5*, *CLB6*, and *BAR1* were constructed by a PCR-based gene-deletion method as previously described [39–41]. The primer sets used in this study are listed in S3 Table. The fragments amplified by PCR were transformed into the wild-type strain and the transformants were selected on the SC medium lacking the corresponding amino acids.

## RNA isolation and quantitative real-time PCR (qRT-PCR)

Yeast cells were pre-cultured overnight in appropriate liquid medium at 28°C. The overnight culture was then diluted to $OD_{600}$ = 0.5 (optical density measured at a wavelength of 600 nm) in fresh medium, and further cultured for 4 hours. Following this, the cells were then collected by centrifuge and total RNA was extracted using ISOGEN reagent (Nippon Gene). From the extract, genomic DNAs were removed using RNeasy Mini kit (Qiagen), and reverse transcription was performed using the Prime Script RT reagent Kit (Takara). The cDNA levels were quantified by qRT-PCR using QuantStudio 5 (Thermo Fisher Scientific) with TB Green Ex Taq (Takara). The primers used for the qRT-PCR were listed in S4 Table. The fold change of the mRNAs was calculated using *SCR1* or *ACT1* as internal control genes and statistically analyzed using Microsoft Office Excel.

## Cell cycle synchronization by α-factor block and release

For the pheromone-induced cell cycle synchronization procedure *MAT***a** *bar1*Δ strains were used to prevent degradation of α-factor and the cell cycle synchronization procedure was followed as previously reported [26]. Yeast cells were pre-cultured overnight in YPD medium at 28°C, then transferred into a fresh YPD medium, and cultured for 4 hours.

After the 4-hour culture, α-factor was added into the culture, and incubated for 2 hours. Following the incubation, after collecting the 0-minute sample, cells were washed with a fresh

YPD medium by centrifuge, transferred into a fresh YPD medium, and incubated at 28˚C. Samples were collected by centrifuge every 10 minutes from the time at release.

## Observation of cell morphology and nuclei

Following the cell cycle arrest using the α-factor as previously discussed, the cells were released and collected at an interval of 20 minutes from the time of release up to 120 minutes. The collected cells were fixed in formaldehyde solution for 1 hour at room temperature. After fixation, cells were washed with 1X Phosphate-buffered saline (PBS) and incubated with 4',6-diamidino-2-phenylindole (DAPI) in a ratio of 1:1 at room temperature. The stained cells were then mounted on glass slides and visualized using a fluorescence microscope (Keyence BZ-X710; Keyence Corporation, Japan) equipped with a 100× oil immersion, under the appropriate excitation and emission settings for DAPI fluorescence. Fluorescent images were acquired using a Keyence BZ-X Viewer software (Keyence Corporation, Japan) and processed with ImageJ version 2.14.0.

## Protein extraction and western-blot analysis

Yeast cells harboring YCplac33-*CLB1*-3HA or YCplac33-*CLB2*-3HA were precultured in SC-Ura medium overnight, and then cells were collected and dissolve into the same amount of YPD medium and cultured for 2 hours. After that, cell cycle synchronization procedures were performed as described above. After releasing, cells (OD600 = 10) were collected every 20 minutes and reacted with sodium hydroxide for protein extraction [42]. Protein samples were loaded onto a 10% SDS-PAGE gel, and electrophoreses were performed. After transferring to a PDVF membrane (Millipore), the membrane was treated with Ponceau and then reacted with the primary antibody, the anti-HA monoclonal antibody HA11, at 4˚C overnight. Reacted with the secondary antibody, the anti-mouse IRDye® 800CW secondary antibodies and IRDye® 680RD secondary antibodies (LI-COR), for 1 hour at room temperature, the Clb1-3HA and Clb2-3HA proteins were visualized and quantified using ODYSSEY CLx (LI-COR).

## Statistical analysis

Microsoft Office Excel was used to perform statistical analyses and generation of graphs. The data was represented as mean ± standard error (SE). Statistical significance was analyzed by One-way ANOVA, followed by Tukey's test. For comparison between two sample groups, t-test was performed. $**P < 0.01$ or $*P < 0.05$ were considered statistically significant.

## Supporting information

**S1 Table. Strains used in this study.**
(DOCX)

**S2 Table. Plasmids used in this study.**
(DOCX)

**S3 Table. Primers used for the gene deletion.**
(DOCX)

**S4 Table. Primers used for the qRT-PCR.**
(DOCX)

**S1 Fig. The cell cycle-regulated expression of *CLB3*, *CLB4*, *CLB5*, and *CLB6* was not significantly changed in the *puf5Δ* mutant.** (A-D) The cell cycle-dependent mRNA levels of *CLB3*,

*CLB4*, *CLB5*, and *CLB6* in the synchronized *bar1Δ* cell (black circle) and *bar1Δ puf5Δ* mutant (grey square). The levels of *CLB3* mRNA (A), *CLB4* mRNA (B), *CLB5* mRNA (C), and *CLB6* mRNA (D) were quantified by qRT-PCR analysis, and the relative mRNA levels were calculated using the *SCR1* reference gene. The vertical axis shows the fold change of mRNA level relative to that in the *bar1Δ* 0 min sample, and the horizontal axis shows the time after release from the G1-phase.
(TIFF)

**S2 Fig. The *puf5Δ clb3Δ* double mutant, the *puf5Δ clb4Δ* double mutant, and the *puf5Δ clb3Δ clb4Δ* triple mutant showed no growth retardation.** The tetrad analysis of the strains that are heterozygous for the alleles of *PUF5*, *CLB3*, and *CLB4*. The cells were sporulated, dissected on a YPD plate, and cultured at 30˚C for 3 days.
(TIFF)

**S3 Fig. The *puf5Δ clb1Δ clb5Δ clb6Δ* quadruple mutant is defective to complete the M-phase.** Morphology and nuclear images of the *clb1Δ clb5Δ clb6Δ* triple mutant (A) and the *puf5Δ clb1Δ clb5Δ clb6Δ* quadruple mutant (B). Cells were synchronously cultured and collected as described in the material and method section. Bright-field (left) and the overlayed (right) were shown. The scale bar represents 2 μm.
(TIFF)

**S4 Fig. The *clb1Δ* mutation did not suppress the lethality of the *ixr1Δ dun1Δ* double mutant.** The tetrad analysis of the strains that are heterozygous for the alleles of *IXR1*, *DUN1*, and *CLB1*. The cells were sporulated, dissected on a YPD plate, and cultured at 30˚C for 3 days.
(TIFF)

**S5 Fig. The expression of *RNR1* was unchanged in the *dun1Δ clb5Δ clb6Δ* triple mutant strain as compared to the *dun1Δ* mutant.** The mRNA levels of *RNR1* in the wild-type strain, the *dun1Δ* mutant, the *clb5Δ* mutant, the *clb6Δ* mutant, the *dun1Δ* mutant, the *dun1Δ clb5Δ* mutant, the *dun1Δ clb6Δ* mutant, the *clb5Δ clb6Δ* mutant, and the *dun1Δ clb5Δ clb6Δ* mutant. The cells were cultured in a YPD medium at 28˚C until the log phase. The *RNR1* mRNA levels were quantified by qRT-PCR analysis, and the relative mRNA levels were calculated using the *ACT1* reference gene. The data shows the mean ± SE (n = 3) of the fold change of *RNR1* in relative to the mRNA level in the wild-type strain. *P < 0.05, **P < 0.01 as determined by Tukey's test. NS indicates no significant change.
(TIFF)

**S6 Fig. The expression of *RNR1* was unchanged in the *puf5Δ clb5Δ clb6Δ* triple mutant.** The mRNA levels of *RNR1* in the wild-type strain, the *puf5Δ* mutant, the *puf5Δ clb5Δ* mutant, the *clb5Δ clb6Δ* mutant, and the *puf5Δ clb5Δ clb6Δ* mutant. The cells were cultured in a YPD medium at 28˚C until the log phase. The *RNR1* mRNA levels were quantified by qRT-PCR analysis, and the relative mRNA levels were calculated using the *ACT1* reference gene. The data shows the mean ± SE (n = 3) of the fold change of *RNR1* relative to the mRNA level in the wild-type strain. NS indicates no significant change.
(TIFF)

**S1 Data.**
(XLSX)

## Acknowledgments

We thank all the members of the Molecular Cell Biology Laboratory for valuable discussions and suggestions.

## Author Contributions

**Conceptualization:** Megumi Sato, Varsha Rana, Kenji Irie.

**Data curation:** Megumi Sato, Varsha Rana, Kenji Irie.

**Formal analysis:** Megumi Sato, Varsha Rana, Kenji Irie.

**Funding acquisition:** Kenji Irie.

**Investigation:** Megumi Sato, Varsha Rana, Kenji Irie.

**Project administration:** Kenji Irie.

**Supervision:** Kenji Irie.

**Validation:** Megumi Sato, Varsha Rana.

**Visualization:** Megumi Sato, Varsha Rana.

**Writing – original draft:** Megumi Sato, Varsha Rana, Kenji Irie.

**Writing – review & editing:** Megumi Sato, Varsha Rana, Yasuyuki Suda, Tomoaki Mizuno, Kenji Irie.

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
