## [Decision Letter · Decision Letter 0]

14 Oct 2024

PONE-D-24-40299The RNA-binding protein Puf5 and the HMGB protein Ixr1 regulate cell cycle-specific expression of CLB1 and CLB2 in Saccharomyces cerevisiaePLOS ONE

Dear Dr. Irie,

Thank you for submitting your manuscript to PLOS ONE. After careful consideration, we feel that it has merit but does not fully meet PLOS ONE’s publication criteria as it currently stands. Therefore, we invite you to submit a revised version of the manuscript that addresses the points raised during the review process.

Both reviewers raise substantial editorial and technical issues and consider that some of your conclusions are not backed up by the experimental evidence that you present. These include the evaluation of the protein expression levels of Clb1 and Clb2 as well as direct measurement of DNA damage. We invite you to submit a revised version of the manuscript that addresses all the points listed below (see the reviewers' reports for more details):

We look forward to receiving your revised manuscript.

Kind regards,

Reiko Sugiura, M.D., PhD.

Academic Editor

PLOS ONE

 Journal requirements: When submitting your revision, we need you to address these additional requirements. 1. Please ensure that your manuscript meets PLOS ONE's style requirements, including those for file naming. The PLOS ONE style templates can be found at https://journals.plos.org/plosone/s/file?id=wjVg/PLOSOne_formatting_sample_main_body.pdf and https://journals.plos.org/plosone/s/file?id=ba62/PLOSOne_formatting_sample_title_authors_affiliations.pdf 2. Thank you for stating the following financial disclosure:  [This research was supported by JSPS KAKENHI Grant Number 22K06074 (to KI).].  Please state what role the funders took in the study.  If the funders had no role, please state: ""The funders had no role in study design, data collection and analysis, decision to publish, or preparation of the manuscript."" If this statement is not correct you must amend it as needed. Please include this amended Role of Funder statement in your cover letter; we will change the online submission form on your behalf. 3. Please expand the acronym “JSPS” (as indicated in your financial disclosure) so that it states the name of your funders in full.This information should be included in your cover letter; we will change the online submission form on your behalf. 4. Thank you for stating the following in the Acknowledgments Section of your manuscript: [We thank all the members of the Molecular Cell Biology Laboratory for valuable discussions and suggestions. This research was supported by JSPS KAKENHI Grant Number 22K06074 (to KI).]We note that you have provided funding information that is not currently declared in your Funding Statement. However, funding information should not appear in the Acknowledgments section or other areas of your manuscript. We will only publish funding information present in the Funding Statement section of the online submission form. Please remove any funding-related text from the manuscript and let us know how you would like to update your Funding Statement. Currently, your Funding Statement reads as follows:  [This research was supported by JSPS KAKENHI Grant Number 22K06074 (to KI).].  Please include your amended statements within your cover letter; we will change the online submission form on your behalf.

Reviewers' comments:

Reviewer's Responses to Questions

**Comments to the Author**

1. Is the manuscript technically sound, and do the data support the conclusions?

Reviewer #1: Yes

Reviewer #2: Partly

2. Has the statistical analysis been performed appropriately and rigorously? 

Reviewer #1: Yes

Reviewer #2: Yes

3. Have the authors made all data underlying the findings in their manuscript fully available?

Reviewer #1: Yes

Reviewer #2: Yes

4. Is the manuscript presented in an intelligible fashion and written in standard English?

Reviewer #1: Yes

Reviewer #2: Yes

5. Review Comments to the Author

Reviewer #1: In this manuscript, the authors reported that the RNA-binding protein Puf5 regulates the expression of CLB1, as previously reported, and also CLB2, albeit to a lesser extent. This regulation of both CLB1 and CLB2 by the Puf5-Ixr1 machinery involves the Dun1 kinase. Additionally, it was observed that the G2-M cyclins, Clb1 and Clb2, have overlapping functions with the S-phase cyclins, Clb5 and Clb6.　

While some potentially interesting results have been observed, some minor corrections are desirable before publication.

1. Please provide the number of experiments conducted or the biological replicates for Figures 3, 4, 5, and 6 in the figure legends.

2. The authors should carefully review the entire text.

Introduction, line 131: The phrase “the arrest of the cell cycle at G2/M phase arrest” contains redundant wording.

Results, line 168: The word "consistently" should be reconsidered. It seems a bit of over-speculation in this context.

Lines 177 and 178: “the expression…CLB5 and CLB6 remained unchanged”. According to Supplemental Figure 1, the expressions of CLB5 and CLB6 were slightly down-regulated, and their expression timing was shifted to later.

Lines 179 and 180: “CLB2 expression was preserved in the puf5Δ mutant”. This contradicts the section title “Expression of CLB2 is reduced in the puf5Δ mutant.”

Lines 209 and 210: “The clb1Δ clb5Δ clb6Δ triple …than the wild-type strain.” There is no wild-type spore in Figure 2B.

Line 213: There is a “-” between the sentences.

Lines 243 and 244: “the puf5Δ clb5Δ double…by the multi-copy CLB2, CLB5, and CLB6”. In Figure 9A, complementation with YEpCLB6 at 37°C was not sufficient.

Lines 269 and 270: “a single copy or…inhibited the growth of the ixr1∆ dun1∆ clb2∆”. The suppression of growth by a single copy CLB2 plasmid was very mild (Figure 10B).

Line 286: The phrase “like the sml1Δ mutation” is a bit confusing, as the authors mentioned earlier in lines 279 and 280 that “the effect by … is different from the effect of the sml1Δ deletion.”

Lines 290: “This is consistent with” sounds too conclusive; it might be better to soften the statement.

Lines 298 and 299: “the dun1Δ clb6Δ double…slow growth”. It is difficult to see the slow-growth phenotype of that mutant (Figure 13A).

Lines 306-310: I think there is a missing “,” in the sentence that begins with “Although a significant...”.

Lines 313: “was may be” is not correct.

Lines 317-319: The last sentence of the results section is purely speculative, so it might be better to move it to the discussion section.

Discussion, line 330: “a clear decrease” should be softened.

Lines 374-381: The sentences may need to be reviewed for clarity regarding the author’s claim.

Reviewer #2: This manuscript by Sato et al. builds upon their previous findings on the regulation of CLB1 expression by the RNA-binding protein Puf5 in yeast genetics. The authors extend their investigation into the broader impact of Puf5 on the cyclins. By using synchronized cell cultures, they reveal that Puf5 regulates not only CLB1 but also CLB2, and suggest that this regulation may be mediated by the HMGB protein Ixr1. Furthermore, they demonstrate that Clb2 can compensate for the functions of Clb5 and Clb6. Their study highlights the significance of the Puf5-Ixr1 mechanism in the fine-tuned regulation of the cell cycle through cyclin gene expression control. However, several issues need to be addressed before publication in PLOS ONE.

<major comments="">

The study proposes a mechanism for Puf5's regulation of the cell cycle by demonstrating the degree of yeast growth and qRT-PCR results using genetics. However, it remains unclear whether the expression levels of cyclin mRNA and cyclin proteins are correlated. To strengthen the validity of the study, it would be helpful to show protein expression levels of Clb1 and Clb2, particularly in Figure 1 and Figures 3C, D at the 60-80 min time points, which are critical to the study.

Lines 218-221: The delay in SIC1 mRNA expression in the puf5Δ clb1Δ clb5Δ clb6Δ quadruple mutant is attributed to a G2 phase delay, but it is also possible that cell cycle synchronization is inconsistent. It would be advisable to soften the expression in the manuscript or support the authors' conclusions by performing nuclear and cell wall staining followed by microscopic observations.

Lines 225-227: Both CLB2 overexpression and PUF5 overexpression suppress the growth defect of the puf5Δ clb1Δ clb5Δ clb6Δ strain, but PUF5 overexpression appears to promote faster growth. Could this be due to PUF5 overexpression regulating target molecules other than CLB2? Alternatively, when using a plasmid, the cell cycle-dependent expression of Clb2 may not be fully replicated, and plasmid-derived Clb2 might not function as effectively as endogenous Clb2. Since PUF5 overexpression regulates endogenous Clb2 expression, it could restore the cell cycle-dependent expression pattern of Clb2, leading to better recovery of Clb2 function than CLB2 overexpression. Current data do not clarify whether CLB2 expressed from a plasmid exhibits cell cycle-dependent expression, making it difficult to determine whether CLB2 overexpression is fully functional. Including clb1Δ clb5Δ clb6Δ-YEplac195 as a control would allow for an evaluation of whether the effects of exogenously expressed CLB2 overexpression are complete or partial.

Lines 265-266: Does CLB1 play a role in the growth defect of ixr1Δ dun1Δ? Would the ixr1Δ dun1Δ clb1Δ clb2Δ strain show improved growth compared to the triple mutant, ixr1Δ dun1Δ clb2Δ? To discuss the functional distinction between Clb1 and Clb2, it would be preferable to show the effects of clb2Δ on ixr1Δ dun1Δ.

Lines 288-291: DNA damage is being indirectly assessed by RNR3 expression. It would be preferable to directly evaluate DNA damage. If not, it may be better to move this statement to the Discussion section. Since there are many repeated sections in the Results and Discussion, combining these sections into one as "Results and Discussion" might be more effective.

Lines 297-299: In Figure 13A, the growth of dun1Δ clb5Δ appears slow, but dun1Δ clb6Δ shows little change in growth compared to dun1Δ and clb6Δ.

Lines 352-353: In Figure 9A, CLB1 overexpression does not appear to suppress the growth defect of puf5Δ clb5Δ. Thus, CLB1 may not compensate for CLB5. It would be important to examine whether CLB1 overexpression can suppress the growth inhibition of puf5Δ clb5Δ clb6Δ. Alternatively, the effect of CLB1 overexpression on the growth and temperature sensitivity of puf5Δ clb5Δ should be more accurately evaluated.

Why can Clb2 compensate for Clb5 and Clb6? A discussion of the physiological significance of this compensation would be desirable.

<minor comments="">

The figures appear somewhat disorganized, so it would be better to reconsider the integration and order of the figures to match the manuscript.

Line 204: The statement that the puf5Δ clb3Δ clb4Δ triple mutant did not show any growth delay could be softened, as Supplemental Figure 2 seems to show a slight growth retardation. A phrase like "showed slight growth retardation" would be preferable.

An explanation of what DUN1 is should be provided not only in the abstract but also in the main text, particularly in the Results section.

The notation of genotypes should be consistent between the manuscript and the figures. For example, in the manuscript, clb1Δ clb5Δ clb6Δ is used, but in Figure 6, clb5Δ clb1Δ clb6Δ is written.

Figures 4A and B are not explained in the manuscript.

Are the cells in Figure 6 in a bar1Δ background? If so, the description in both the figure and the manuscript should be revised accordingly.</minor></major>

6. PLOS authors have the option to publish the peer review history of their article (what does this mean?). If published, this will include your full peer review and any attached files.

Reviewer #1: No

Reviewer #2: No

---

## [Author Response · Author response to Decision Letter 0]

26 Nov 2024

Our responses to Journal requirements

The revise paper was adapted to the PLOS ONE style.

 [This research was supported by JSPS KAKENHI Grant Number 22K06074 (to KI).]. 

I filled out the cover letter.

3. Please expand the acronym “JSPS” (as indicated in your financial disclosure) so that it states the name of your funders in full.

I filled out the cover letter.

[We thank all the members of the Molecular Cell Biology Laboratory for valuable discussions and suggestions. This research was supported by JSPS KAKENHI Grant Number 22K06074 (to KI).]

 [This research was supported by JSPS KAKENHI Grant Number 22K06074 (to KI).]. 

The description has been removed from the text.

Our responses to the reviewers’ comments and changes in the revised manuscript

Title : The RNA-binding protein Puf5 and the HMGB protein Ixr1 regulate cell cycle-specific expression of CLB1 and CLB2 in Saccharomyces cerevisiae

Authors: Megumi Sato, Varsha Rana, Yasuyuki Suda, Tomoaki Mizuno, Kenji Irie

Reviewer #1

1. Please provide the number of experiments conducted or the biological replicates for Figures 3, 4, 5, and 6 in the figure legends.

 For all cell cycle experiments, we performed the experiments two times and presented the representative data. As for the experiment for Figure 2 in the revised manuscript (Figure 3 in the first edition), we isolated RNA from the same culture for Figure 3A and observed similar expression patterns to Figures 2A-2D by qRT-PCR analysis. In addition, for each experiment for Figures 5, 6, and 8 in the revised manuscript (Figures 4, 5, and 6 in the first edition, respectively), we repeated the same experiment and obtained similar results. Therefore, we presented the representative data for each figure and noted in the Figure legend. 

2. The authors should carefully review the entire text.

 According to the reviewer’s comment, we thoroughly reviewed and re-wrote the entire manuscript. 

Lines 177 and 178: “the expression…CLB5 and CLB6 remained unchanged”. According to Supplemental Figure 1, the expressions of CLB5 and CLB6 were slightly down-regulated, and their expression timing was shifted to later.

 The expression of CLB5 and CLB6 was actually decreased slightly in the bar1∆ puf5∆ double mutant than that in the bar1∆ mutant as reviewer pointed. However, the extent was much lower than the downregulation of CLB1 and CLB2 expressions. Therefore, we considered the effect on CLB1/CLB2 significant, but not on CLB5/CLB6. We mentioned the results in lanes 175-179, page 9, of the revised manuscript. 

Lines 243 and 244: “the puf5Δ clb5Δ double…by the multi-copy CLB2, CLB5, and CLB6”. In Figure 9A, complementation with YEpCLB6 at 37°C was not sufficient.

 In Figure 10A in the revised figures, the puf5∆ clb5∆ double mutant strain harboring YEpCLB6 grew slightly better than the strain with a vector at 37℃, but this suppression was far weaker than CLB5 or CLB2. Therefore, from these results, we assume that the effect of the multi-copy CLB6 is far milder than that of the multi-copy CLB2 or CLB5. We included the detailed explanation of these results in lanes 290-294, page 14, in the revised manuscript.

Lines 269 and 270: “a single copy or…inhibited the growth of the ixr1∆ dun1∆ clb2∆”. The suppression of growth by a single copy CLB2 plasmid was very mild (Figure 10B).

 As for the inhibitory effect of a single-copy CLB2, we carefully interpret the results and concluded that the effect was milder than that of a multi-copy CLB2, and that controlling the dosage of Clb2 was essentially important for the cell survival in the dun1∆ background. We described the results in lanes 330-335, page 15, and discussed the dosage effect of Clb2 in lanes 463-470, page 21, in the discussion section of the revised manuscript. 

Lines 298 and 299: “the dun1Δ clb6Δ double…slow growth”. It is difficult to see the slow-growth phenotype of that mutant (Figure 13A).

 We cautiously interpreted the data and reasoned out that the dun1∆ clb6∆ double mutant did not show growth retardation. Reflecting that, we revised the manuscript and described the data in lanes 376-378, page 17.

Lines 317-319: The last sentence of the results section is purely speculative, so it might be better to move it to the discussion section.

 In response to the reviewer’s suggestion, we moved the sentences mentioning to the relationship between RNR3 expressions and DNA damage into discussion section. We discussed the hypothesis about DNA damage in lanes 483-509, pages 22-23.

Lines 374-381: The sentences may need to be reviewed for clarity regarding the author’s claim.

 According to the revision of the manuscript, we re-wrote whole sentences in the discussion section carefully considering our data.

Reviewer #2

The study proposes a mechanism for Puf5's regulation of the cell cycle by demonstrating the degree of yeast growth and qRT-PCR results using genetics. However, it remains unclear whether the expression levels of cyclin mRNA and cyclin proteins are correlated. To strengthen the validity of the study, it would be helpful to show protein expression levels of Clb1 and Clb2, particularly in Figure 1 and Figures 3C, D at the 60-80 min time points, which are critical to the study.

 According to the reviewer’s suggestion, we examined the 3HA-tagged Clb1 and Clb2 protein levels in the bar1∆ mutant or the bar1∆ puf5∆ double mutant throughout cell cycle. For this experiment, we transformed a single-copy CLB1-3HA or a CLB2-3HA plasmid into the two strains and collected samples as described in the material and method section. Consequently, as shown in Figure 3, we observed a significant decrease in the Clb1 protein levels in the bar1∆ puf5∆ double mutant than that in the bar1∆ mutant, even though the basal Cl1 protein levels were low. Therefore, the downregulation of CLB1 mRNA levels seem to be reflected in the protein levels. As for the Clb2, the protein level was mildly decreased at 40 min time point in the bar1∆ puf5∆ double mutant compared to the bar1∆ mutant. However, the Clb2 protein levels did not show significant difference between the two mutants after 60 min time point. Nevertheless, regarding the data at 40 min time point, we assume that we could confirm the delay in the induction of Clb2 protein expression in the puf5∆ strains. We described the results in lanes 154-170, page 8, in the revised manuscript. 

 For one reason why Clb2 protein levels were not significantly decreased in the puf5∆ strains as mRNA level, we hypothesize that there are some compensatory mechanisms for the Clb2 protein levels, since Clb2 is the most important mitotic cyclin. We described this in lanes 164-167, page 8, in the revised manuscript. 

 Nevertheless, it is certain that the downregulation of CLB2 expression in the puf5∆ strains, albeit very mild, physiologically important in the deficiency of S-phase cyclins. Therefore, we regard the observed delay in the Clb2 induction to be significant and important for cell growth.

Lines 218-221: The delay in SIC1 mRNA expression in the puf5Δ clb1Δ clb5Δ clb6Δ quadruple mutant is attributed to a G2 phase delay, but it is also possible that cell cycle synchronization is inconsistent. It would be advisable to soften the expression in the manuscript or support the authors' conclusions by performing nuclear and cell wall staining followed by microscopic observations.

 We observed the morphology and nuclei of the clb1∆ clb5∆ clb6∆ triple mutant and the puf5∆ clb1∆ clb5∆ clb6∆ quadruple mutant after releasing from G1-phase arrest using nuclear staining by DAPI. In both mutants, cell cycle progression was successfully observed from morphological changes and nuclei behaviors, even though the puf5∆ clb1∆ clb5∆ clb6∆ quadruple mutant showed highly elongated phenotype. We also observed that the puf5∆ clb1∆ clb5∆ clb6∆ quadruple mutant was not able to complete M-phase. Therefore, consistent with the data in Figure 8C that SIC1 expression was not induced in the quadruple mutant, the mutant was suggested to have difficulties in accomplishing the mitosis. We presented microscopic data in Figure S3A and S3B and mentioned in lanes 251-266, pages 12-13, in the revised manuscript.

Lines 225-227: Both CLB2 overexpression and PUF5 overexpression suppress the growth defect of the puf5Δ clb1Δ clb5Δ clb6Δ strain, but PUF5 overexpression appears to promote faster growth. Could this be due to PUF5 overexpression regulating target molecules other than CLB2? Alternatively, when using a plasmid, the cell cycle-dependent expression of Clb2 may not be fully replicated, and plasmid-derived Clb2 might not function as effectively as endogenous Clb2. Since PUF5 overexpression regulates endogenous Clb2 expression, it could restore the cell cycle-dependent expression pattern of Clb2, leading to better recovery of Clb2 function than CLB2 overexpression. Current data do not clarify whether CLB2 expressed from a plasmid exhibits cell cycle-dependent expression, making it difficult to determine whether CLB2 overexpression is fully functional. Including clb1Δ clb5Δ clb6Δ-YEplac195 as a control would allow for an evaluation of whether the effects of exogenously expressed CLB2 overexpression are complete or partial.

 In the experiment for Figure 3B, strains harboring a YCplac33-CLB2-3HA plasmid showed rhythmical expression patterns of Clb2-3HA protein. Therefore, we assume that the plasmid-induced expression of CLB2 is also regulated in a cell cycle specific manner. Actually, multi-copy PUF5 suppressed the growth defect of the puf5∆ clb1∆ clb5∆ clb6∆ quadruple mutant stronger than multi-copy-CLB2, but we do not assume that these results originated from the difference of expressional patterns between plasmid-induced CLB2 and endogenous CLB2. Rather, considering that Puf5 is a multi-functional RNA binding protein, we hypothesize that the regulation of other target genes by Puf5 contributed to the phenotype. We discussed this suggestion in lanes 428-433, page 20, in the revised manuscript.

Lines 265-266: Does CLB1 play a role in the growth defect of ixr1Δ dun1Δ? Would the ixr1Δ dun1Δ clb1Δ clb2Δ strain show improved growth compared to the triple mutant, ixr1Δ dun1Δ clb2Δ? To discuss the functional distinction between Clb1 and Clb2, it would be preferable to show the effects of clb2Δ on ixr1Δ dun1Δ.

 For clarifying the contribution of CLB1 expression to the growth defect of the dun1∆ ixr1∆ double mutant, we conducted the tetrad analysis of the strains heterozygous for IXR1, DUN1, and CLB1 genes and presented the data in Figure S4. The clb1∆ mutation was not able to suppress the lethality of the dun1∆ ixr1∆ double mutant. Thus, we conclude that the increased expression of CLB1 is not the cause of the lethality. Regarding the data, the physiological significance of the control of CLB1 or CLB2 by Ixr1 may be different. The description of the tetrad analysis results was included in lanes 327-329, page 15, and we discussed the results in the discussion section of the revised manuscript.

Lines 288-291: DNA damage is being indirectly assessed by RNR3 expression. It would be preferable to directly evaluate DNA damage. If not, it may be better to move this statement to the Discussion section. Since there are many repeated sections in the Results and Discussion, combining these sections into one as "Results and Discussion" might be more effective.

According to the reviewer’s suggestion, we moved the sentences into discussion section about the speculation of the occurrence of DNA damage response from RNR3 expressions. Moreover, we only described objective data and evoked hypothesis in the results section, and all subjective interpretations were moved to discussion section.

Lines 297-299: In Figure 13A, the growth of dun1Δ clb5Δ appears slow, but dun1Δ clb6Δ shows little change in growth compared to dun1Δ and clb6Δ.

We carefully evaluated the tetrad analysis results and concluded that the dun1∆ clb6∆ double mutant grew as well as the wild-type strain. The results were described in lanes 376-378, page 17, of the revised manuscript.

Lines 352-353: In Figure 9A, CLB1 overexpression does not appear to suppress the growth defect of puf5Δ clb5Δ. Thus, CLB1 may not compensate for CLB5. It would be important to examine whether CLB1 overexpression can suppress the growth inhibition of puf5Δ clb5Δ clb6Δ. Alternatively, the effect of CLB1 overexpression on the growth and temperature sensitivity of puf5Δ clb5Δ should be more accurately evaluated.

 According to reviewer’s point, we discreetly evaluated the results shown in revised Figure 10A and reasoned out that the multi-copy CLB2 did not suppress the growth defect of the puf5∆ clb5∆ double mutant. From these results, Clb1 seems not to function as a substitute for Clb5 as reviewer pointed. We described the interpretation of the results in lanes 295-302, page 14, of the revised manuscript.

Why can Clb2 compensate for Clb5 and Clb6? A discussion of the physiological significance of this compensation would be desirable.

 According to reviewer’s suggestion, we discussed our hypothesis why overexpression of G2/M-phase cyclin Clb2 can compensate for Clb5 in lanes 435-451, page 20, of the revised manuscript.

The figures appear somewhat disorganized, so it would be better to reconsider the integration and order of the figures to match the manuscript.

 We carefully arranged the figures in the order in which they are mentioned in the revised manuscript.

Line 204: The statement that the puf5Δ clb3Δ clb4Δ triple mutant did not show any growth delay could be softened, as Supplemental Figure 2 seems to show a slight growth retardation. A phrase like "showed slight growth retardation" would be preferable.

 In line with the reviewer’s indication, we reinterpreted the tetrad analysis data. We presented the results in the revised Figure S2 as “the puf5∆ clb3∆ clb4∆ triple mutant showed only a slight growth retardation” in lanes 224-227, page 11, of the revised manuscript. Since the puf5∆ mutation caused much severe growth defect in the clb5∆ clb6∆ double mutation background than in the clb3∆ clb4∆ condition, we assume that the CLB2 expression is vital for cell survival in the deficiency of S-phase cyclins. 

An explanation of what DUN1 is should be provided not only in the abstract but also in the main text, particularly in the Results section.

 We included the explanation of Dun1 in lanes 314-319, page 15, in the results section of the revised manuscript.

The notation of genotypes should be consistent between the manuscript and the figures. For example, in the manuscript, clb1Δ clb5Δ clb6Δ is used, but in Figure 6, clb5Δ clb1Δ clb6Δ is written.

 In response to reviewer’s pointing, we thoroughly checked all of the notation of strains in the manuscript and figures and revised inconsistent notions.

Figures 4A and B are not explained in the manuscript.

 We thoroughly checked the manuscript and explained all shown data in the r

---

## [Decision Letter · Decision Letter 1]

11 Dec 2024

The RNA-binding protein Puf5 and the HMGB protein Ixr1 regulate cell cycle-specific expression of CLB1 and CLB2 in Saccharomyces cerevisiae

PONE-D-24-40299R1

Dear Dr. Irie,

We’re pleased to inform you that your manuscript has been judged scientifically suitable for publication and will be formally accepted for publication once it meets all outstanding technical requirements.

Kind regards,

Reiko Sugiura, M.D., PhD.

Academic Editor

PLOS ONE

Additional Editor Comments (optional):

Reviewers' comments:

Reviewer's Responses to Questions

**Comments to the Author**

1. If the authors have adequately addressed your comments raised in a previous round of review and you feel that this manuscript is now acceptable for publication, you may indicate that here to bypass the “Comments to the Author” section, enter your conflict of interest statement in the “Confidential to Editor” section, and submit your "Accept" recommendation.

Reviewer #1: All comments have been addressed

Reviewer #2: All comments have been addressed

2. Is the manuscript technically sound, and do the data support the conclusions?

Reviewer #1: Yes

Reviewer #2: Yes

3. Has the statistical analysis been performed appropriately and rigorously? 

Reviewer #1: Yes

Reviewer #2: Yes

4. Have the authors made all data underlying the findings in their manuscript fully available?

Reviewer #1: Yes

Reviewer #2: Yes

5. Is the manuscript presented in an intelligible fashion and written in standard English?

Reviewer #1: Yes

Reviewer #2: Yes

6. Review Comments to the Author

Reviewer #1: The authors have addressed all the points raised by this reviewer in the initial version of the manuscript.

Reviewer #2: The authors have responded to the reviewers' comments with integrity, conducted appropriate additional experiments, and provided clear explanations. The revised manuscript is deemed to have sufficient merit for publication in PLOS ONE.

7. PLOS authors have the option to publish the peer review history of their article (what does this mean?). If published, this will include your full peer review and any attached files.

Reviewer #1: No

Reviewer #2: **Yes: **Ryosuke Satoh

---

## [Editor Report · Acceptance letter]

17 Dec 2024

PONE-D-24-40299R1 

PLOS ONE

Dear Dr. Irie, 

I'm pleased to inform you that your manuscript has been deemed suitable for publication in PLOS ONE. Congratulations! Your manuscript is now being handed over to our production team.

Kind regards, 

on behalf of

Dr. Reiko Sugiura 

Academic Editor

PLOS ONE